# Vertical distribution of ice nucleating particles over the boreal forest of Hyytiälä, Finland

Zoé Brasseur[1], Julia Schneider[2], Janne Lampilahti[1], Ville Vakkari[3,4], Victoria A. Sinclair[1], Christina J. Williamson[1,3], Carlton Xavier[5,6], Dmitri Moisseev[1,3], Markus Hartmann[7], Pyry Poutanen[1], Markus Lampimäki[1], Markku Kulmala[1], Tuukka Petäjä[1], Katrianne Lehtipalo[1,3], Erik S. Thomson[8], Kristina Höhler[2], Ottmar Möhler[2], and Jonathan Duplissy[1]

[1]Institute for Atmospheric and Earth System Research/Physics, Faculty of Science, University of Helsinki, Helsinki, Finland
[2]Institute of Meteorology and Climate Research, Karlsruhe Institute of Technology, Karlsruhe, Germany
[3]Finnish Meteorological Institute, Helsinki, Finland
[4]Atmospheric Chemistry Research Group, Chemical Resource Beneficiation, North-West University, Potchefstroom, South Africa
[5]Department of Physics, Lund University, Lund, Sweden
[6]Swedish Meteorological and Hydrological Institute (SMHI), Norrköping, Sweden
[7]Leibniz Institute for Tropospheric Research, Atmospheric Microphysics, Leipzig, Germany
[8]Department of Chemistry and Molecular Biology, Atmospheric Science, University of Gothenburg, Gothenburg, Sweden

*Correspondence to*: Zoé Brasseur (zoe.brasseur@helsinki.fi)

**Abstract.**

Ice nucleating particles (INPs) play a crucial role in initiating ice crystal formation in clouds, influencing the dynamics and optical properties of clouds and their impacts on precipitation and the climate system. Despite their importance, there is limited knowledge about the vertical distribution of INPs. This study focuses on aircraft measurements conducted during spring 2018 above the boreal forest of Hyytiälä, Finland. Similarities between INP concentrations, activated fractions, particle concentrations and size distributions observed at ground level and in the boundary layer aloft indicate that surface particles and INPs are efficiently transported and mixed within the boundary layer. INP concentrations observed in the boundary layer are best predicted by a parameterization describing near-surface INP concentrations driven by the abundance of biogenic aerosol in the Finnish boreal forest, suggesting that biogenic INPs are dominant in the boundary layer above the same environment. Most of the INP concentrations and activated fractions observed in the free troposphere are notably lower than in the boundary layer, and the distinct particle size distributions suggest that different aerosol populations, likely resulting from long-range transport, are present in the free troposphere. However, we show one case where higher INP concentrations are observed in the free troposphere and where a homogeneous particle population exists from the surface to the free troposphere. This indicates that surface particles and INPs from the boreal forest can occasionally reach the free troposphere, which is particularly important as the INPs in the free troposphere can further travel horizontally and/or vertically and impact cloud formation.

# 1 Introduction

Clouds are a key element of the Earth's climate system because they influence the hydrological cycle and the Earth's radiative budget. However, cloud processes, and especially the interactions between aerosols and clouds, remain highly uncertain in weather forecasting and climate projections (Forster et al., 2021). Ice nucleating particles (INPs) are a rare subset of atmospheric aerosol particles which can trigger the formation of ice crystals in clouds (Pruppacher and Klett, 2010). INPs can influence precipitation, cloud microphysical and optical properties, and the lifetime of clouds (Hoose and Möhler, 2012), and thus strongly influence the Earth's radiative balance. However, the mechanisms responsible for ice formation and evolution in clouds are poorly understood, partly due to our lack of knowledge concerning the identity, sources, abundance, transport patterns and therefore global spatial distribution of INPs in the atmosphere (Murray et al., 2021). The sources of INPs in the atmosphere are complex and include natural sources, such as land and ocean emissions, as well as anthropogenic sources such as agricultural and industrial activities and biomass burning. INPs from different sources may exhibit distinct ice nucleation activities due to differences in their chemical compositions, sizes, phases, and morphologies (Kanji et al., 2017). For example, desert dust is one of the most important sources of atmospheric INPs active at temperatures below -15 °C (Hoose and Möhler, 2012; Kanji et al., 2017; Sanchez-Marroquin et al., 2023; Vergara-Temprado et al., 2017). Biological aerosols are considered another widely present type of INPs (Dreischmeier et al., 2017; Morris et al., 2004; O'Sullivan et al., 2018; O'Sullivan et al., 2015; Wex et al., 2019). Although their global emissions are lower than dust, they can form ice at relatively warmer temperatures depending on the nature of the bioaerosols (Després et al., 2012). For example, the bacteria *Pseudomonas Syringae* is a very efficient INP at temperatures as warm as -2 °C (Joly et al., 2013; Maki et al., 1974). In addition, biological particles, including bacteria, have been found in dust aerosols, possibly enhancing their ice nucleation activity (Barr et al., 2023; Conen et al., 2011; Meola et al., 2015). To overcome our lack of knowledge concerning INPs, there is a need for more observations of INPs worldwide. Such measurements are also needed to develop accurate parameterizations, which are an important tool used to constrain heterogeneous ice nucleation predictions in models (e.g., DeMott et al., 2010; Fletcher, 1962; Meyers et al., 1992).

Over the past decades, a large number of INP field observations have been carried out at ground level around the world (e.g., Belosi et al., 2014; Schrod et al., 2020; Welti et al., 2020), with fewer studies conducted at higher altitudes in the atmosphere (e.g., Rogers et al., 2001a, b; DeMott et al., 2003a; Lacher, 2018). However, given that clouds form at high altitudes in the natural environment, conducting INP measurements there and investigating the vertical distribution of INPs in the atmosphere is crucial. There has been no consistent conclusion on the vertical distribution of INPs in the atmosphere so far, partly because such distribution varies greatly depending on several factors such as orography, underlying surface, local sources and sinks of INPs, influence of long-range transport of particles, and overall atmospheric stratification and weather conditions. For example, Patade et al. (2014) reported that INP concentrations measured over India during the monsoon season were the highest over inland continental regions, and that the concentrations generally decreased with altitude in response to decreasing aerosol particle concentrations. Vychuzhanina et al. (1988; 1996) showed that INP concentrations measured over Eastern

Europe generally decreased with height and that concentrations measured below 4 km were essentially dependent on the type of underlying surface and the presence of local sources of pollution. Twohy et al. (2016) reported that INP concentrations measured in the boundary layer over a forested site in western US were about the same or slightly lower than concentrations observed at ground level and at the top of the forest canopy, while INP concentrations measured primarily in the free troposphere were much lower. Such decrease in INP concentrations was linked to decreasing fluorescent biological aerosol particle and total particle concentrations, suggesting that the canopy was likely the source of INPs. Seifried et al. (2021) sampled INPs above the canopy of a birch forest in the Alps of Upper Austria using a drone and found that the INP concentrations were significantly lower compared to ground-level samples, concluding that the INP emitted from the forest vegetation were diluted in the ambient air when transported above the forest canopy. On the other hand, DeMott et al. (2003), Stith et al. (2009) and Schrod et al. (2017) observed increased INP concentrations in elevated layers due to the presence of dust plumes, and concluded that transported dust could be a major source of INPs in the troposphere. He et al. (2023) showed how a cold front passage introduced aged or coated mineral dust INPs in the troposphere, leading to increased INP concentrations at relatively high altitudes (4-5 km), while INPs were mostly concentrated in the boundary layer before the cold front passage. Similarly, Levin et al. (2019) observed an increase in INP concentration and in the fraction of total aerosol particles capable of ice nucleation from the surface up to approximately 7 km above sea level in wintertime in California, and suggested that pollution aerosols near the surface were poor sources of INPs. Aircraft observations carried out in China reported that INP concentration generally decreased with height, although larger particles ($> 0.5$ µm) present in the upper troposphere, which were likely dust particles transported from distant deserts, exhibited better ice-nucleating abilities compared to those near the surface (He et al., 2021). Some studies show no clear trend(s) in the vertical distribution of INPs concentrations (Hobbs and Deepak, 1981; Rogers et al., 2001b; Rosinski, 1967). Prenni et al. (2009) conducted airborne measurements in northern Alaska and found that INP concentrations were generally higher above the boundary layer and were likely influenced by long-range transport. However, they also show some cases with increased INP concentrations within the boundary layer and concluded that local and regional sources were then contributing more to the measured INPs. Overall, these varying results indicate that the vertical distribution of INPs is sometimes closely related to underlying surface conditions, while, in other instances, long-range transport of particles seems to dominate. Overall, the vertical distribution of INPs highly depends on the environment where the measurements are conducted, and therefore it is important to investigate the vertical distribution over various environments, especially over those that have been understudied in the past.

Boreal forests constitute one such underrepresented environment, and very little is known concerning the vertical distribution of INPs over this environment. Boreal forests represent one-third of all forested land and cover 15 million square kilometers of land (Tunved et al., 2006). They are primarily located in the Arctic and sub-Arctic regions of the continental Northern Hemisphere, and are therefore generally far from anthropogenic and dust sources. Boreal forests are characterized by high concentrations of biogenic aerosol (Kulmala et al., 2013; Tunved et al., 2006) and their vegetation is among the strongest emitters of primary biological aerosol particles (Després et al., 2012). A recent study from Schneider et al. (2021) showed that Finnish boreal forests are also an important source of biogenic INPs, which may contribute substantially to the total INP

population in such environment. These results agree well with previous studies conducted in similar forested environments. Prenni et al. (2009b) for example showed that INP concentrations and abundance in a pristine rainforest of the Amazon basin could be partly explained by local emissions of biological particles. Huffman et al. (2013), who performed measurement in a semi-arid pine forest of North America, found a strong correlation between fluorescent biological particles and INPs during rain events. Similarly, results presented in Prenni et al. (2013) suggest that biological particles represent a significant portion of rain-generated INPs measured at a forested site in Colorado. Tobo et al. (2013) conducted measurements in a midlatitude ponderosa pine forest ecosystem in Colorado and found significant correlations between INP concentrations and the concentration of ambient fluorescent biological aerosol particles. Finally, Iwata et al. (2019) carried out measurements near forested mountain slopes in Japan and found that biological particles played an important role as INPs for temperatures warmer than -22 °C, especially during rainfall events. However, all these observations were carried out at ground level and did not examine the transport of such INPs to higher altitudes. In addition, to our knowledge, no INP measurements have been conducted above a boreal forest environment. The aforementioned study from Seifried et al. (2021) was conducted in an alpine forest with similar vegetation to boreal forests, but their observations were limited to an altitude of 45 m. The importance of boreal forests as a source of INPs, together with the lack of knowledge concerning the overall vertical distribution of INPs above this environment, emphasize the need for measurements at higher altitudes in these specific regions.

In this study, we present filter-based measurements of INPs conducted at ground level and aloft in the boundary layer and free troposphere (up to an altitude of 3.5 km) in and above a Finnish boreal forest during spring 2018. The measurements were organized in the framework of a larger ice nucleation measurement campaign, called HyICE-2018, which took place at the Station for Measuring Ecosystem–Atmosphere Relations (SMEAR II; Hari and Kulmala, 2005) in Hyytiälä, Finland and is presented in details in Brasseur et al. (2022). Results from the HyICE-2018 are also available from Paramonov et al. (2020), who presented ground-based INP concentrations measured with the Portable Ice Nucleation Chamber (PINC) during the first part of the campaign. The study from Schneider et al. (2021)extended their measurements for more than one year after the HyICE-2018 campaign and focused on immersion freezing INPs measured with the Ice Nucleation Spectrometer of the Karlsruhe Institute of Technology (INSEKT). They showed that the surface INP concentrations have a clear seasonal cycle that appears linked to the abundance of boreal biogenic aerosol. Finally, Vogel et al. (2024) presented ground-based measurements conducted with the Portable Ice Nucleation Experiment (PINE) below -24 °C and found moderate correlations between INP concentrations and concentrations of particles larger than 0.5 µm as well as concentrations of fluorescent aerosol particles, hinting at a possible biological source of INPs active below -24 °C. Building from these previously published results, the objective of this study is to describe the vertical variability in INP concentrations from ground level to the free troposphere above the Finnish boreal forest environment. To do so, we use instrumentation installed both onboard a measurement aircraft and at the SMEAR II measurement site, which allows for comparison between INP measurements and simultaneous measurements of many particle and meteorological variables.

## 2 Methods

The data presented here was collected during an aircraft measurement campaign organized in spring 2018 above the boreal forest at the SMEAR II station in Hyytiälä, southern Finland (61°51′ N, 24°17′ E; 181 m above sea level; Fig. A1). Data from 135 19 flights conducted between 20 April and 19 May 2018 are presented together with continuous ground-based measurements from SMEAR II.

### 2.1 Overview of the flight measurements

The airborne measurements were conducted onboard a Cessna 172 aircraft, and each flight started and ended at the Tampere-Pirkkala airport (61°25′ N, 23°35′ E, 119 m above sea level; Fig. A1) located approximately 60 km south-west from SMEAR 140 II (Fig. 1a). A typical flight lasted approximately 3 hours and consisted of a 30-minute transit to the measurement area above SMEAR II at an altitude of 300 m above ground level (a.g.l.) followed by a single vertical profile from 300 to 3500 m a.g.l realized over approximately 20-kilometer-long segments above the measurement site, as shown in Fig. 1a. In this way, the measurements covered the boundary layer and the lowest part of the free troposphere (Fig. 2). Profiles were always flown perpendicular to the mean wind direction to avoid contamination from the airplane's engine exhaust. The airspeed was kept at 145 130 km.h$^{-1}$ during the measurement flights.

The majority of instruments were built into a rack located behind the front row seats. The instruments were supplied with sample air collected through an inlet mounted outside the aircraft. The inlet's design was adopted from the University of Hawaii's shrouded solid diffuser inlet originally presented in McNaughton et al. (2007) for use aboard a DC-8 aircraft. The sample air was transported to the instruments inside the aircraft's cabin through a stainless-steel tube (22 mm inner diameter), 150 and the exhaust air exited through a venturi mounted on the right main gear leg. The forward movement of the aircraft during the flight together with the suction from the venturi provided the necessary sample air flow.

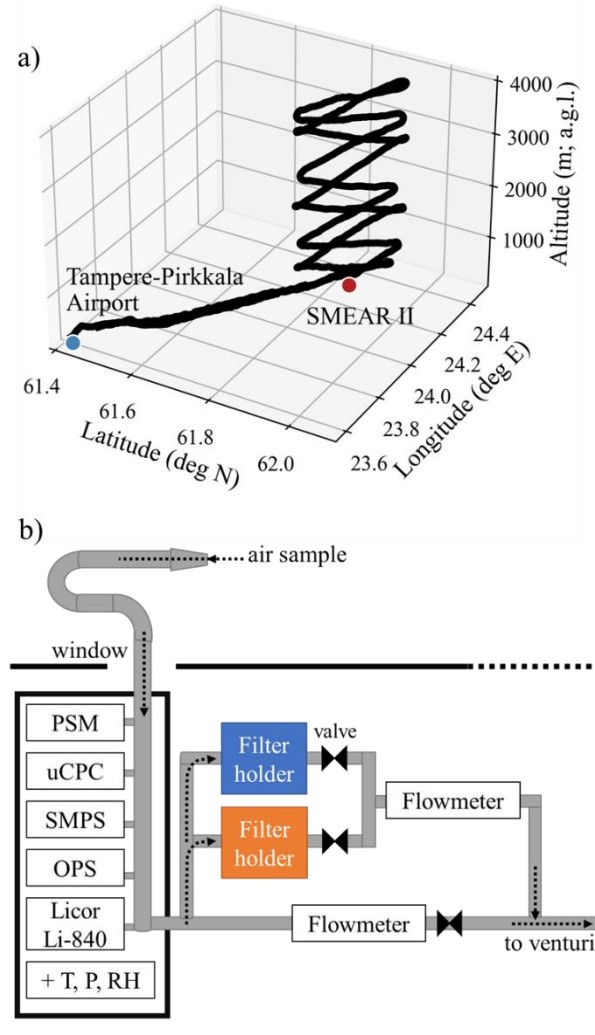

**Figure 1: a) Example of a flight track from Tampere-Pirkkala airport to SMEAR II. The distance from the airport to the station is approximately 60 km. The location of SMEAR II with respect to Northern Europe is given in Fig. A1. b) Schematic of the instrumental setup viewed from above inside the Cessna 172, described in detail in Section 2.1.1.**

### 2.1.1 Particle measurements

In flight monitoring of aerosol microphysical properties was conducted using a particle size magnifier (PSM; Airmodus model A10) operated with a condensation particle counter (CPC; TSI model 3010) measuring the $> 1.5$ nm particle number concentration at a 1 s time resolution, an ultrafine CPC (TSI model 3776) measuring the $> 3$ nm particle number concentration at a 1 s time resolution, a custom-built scanning mobility particle sizer (SMPS) comprised of a short Hauke-type differential mobility analyzer (DMA) and a CPC (TSI model 3010) measuring the aerosol number size distribution in the size range of 10-400 nm at a time resolution of approximately 2.2 minutes, and an optical particle sizer (OPS; TSI model 3330) measuring the aerosol number size distribution in the size range of 0.3-10 µm. The shrouded solid diffuser inlet used has a 5.0 µm

aerodynamic diameter cutoff, thus particle concentrations and number size distributions from 1.5 nm to 5.0 µm were measured

with this setup. The flow rate going through the main sampling line was recorded using a flow meter (TSI, model 4000) and adjusted manually using a valve (Fig 1b, bottom flow meter) to keep it constant at 47 L min$^{-1}$. The instruments drew air from the main sampling line using core sampling inlets.

In addition, meteorological data (relative humidity, temperature, and pressure) were measured with a Rotronic HygroClip-S and a PT1-100 temperature sensor, and water vapor concentration was measured with a LI-COR Li-840 gas analyzer. The

170 aircraft's GPS receiver also recorded latitude, longitude, and flight altitude. Additional information concerning the instrumentation and the layout used in the Cessna 172 can be found in Schobesberger et al. (2013), Väänänen et al. (2016), Leino et al. (2019) and Lampilahti et al. (2021).

### 2.1.2 INP filter sampling and analysis with the INSEKT

To determine the INP concentration in the ambient air, aerosol particles were collected on 47 mm Whatman nucleopore track-

175 etched polycarbonate membrane filters with a pore size of 0.2 µm. Before sampling, the filters were pre-cleaned by soaking them with 10 % $H_2O_2$ for 10 minutes. Afterwards, they were rinsed three times with deionized water that was passed through a 0.1 µm Whatman syringe filter. After drying the prepared filters, they were placed in filter holders made of stainless steel. For each flight, two filter holders were connected to the sampling line onboard the aircraft, as shown in Fig. 1b, and the objective was to sample one filter in the boundary layer and the other filter in the free troposphere. The boundary-layer depth

was estimated during the flights using the real-time particle concentration, water vapor concentration and potential temperature monitoring, and ranged between 500 and 2500 m approximately. More information concerning the estimation of the boundary-layer depth is given in section 2.3. A third filter holder was installed at SMEAR II to sample ambient aerosol particles at ground level for the same duration as the flight (≈3 hours).

During the flight, both sampling lines going to the filter holders were kept closed until the aircraft was approximately 30 km

from Tampere to avoid urban contamination. Then the sampling lines were opened and closed alternately, depending on the atmospheric layer sampled. The volumetric flow rate going through the filters was recorded (Fig.1b, top flow meter) and kept at the highest rate possible while maintaining the main flow rate at 47 L min$^{-1}$. The average flow rate going through the filter sampled in the boundary layer was 9 L min$^{-1}$ with an average sampling time of 70 minutes, while the average flow rate going through the filter sampled in the free troposphere was 7 L min$^{-1}$ with an average sampling time of 1 hour. At SMEAR II, the

ground-level filter was sampled from a measurement container using a vertical sampling line connected to a total aerosol inlet. The inlet height was approximately 4 m a.g.l. and the average flow rate through the filter was 15 L min$^{-1}$ with an average sampling time of 3 hours. After sampling, the filters were placed in sterile petri dishes which were wrapped in aluminum foil and stored frozen until the samples were analyzed for their INP content (typically within a week after the sampling).

To analyze the INP content of the collected aerosol samples, the INSEKT instrument was used. The INSEKT is based on an

195 ice spectrometer developed at the Colorado State University (Hill et al., 2016) and is described in more detail in Schiebel (2017). With the INSEKT, INP concentration are measured as a function of the activation temperature in the immersion

freezing mode between -5 and - 26 °C. The INP analysis applied to the aerosol samples collected for this study mostly followed the experimental procedure described in Schneider et al. (2021). First, we used Milli-Q purified water (18.2 MΩ.cm), which was passed through a 0.1 µm Whatmann syringe filter to remove possible remaining impurities, to wash the sampled aerosol particles from the filter membrane into a solution. As the sampling times on board the Cessna aircraft were shorter than those in the study of Schneider et al. (2021), the INP content on each collected filter was expected to be lower. For this reason, and to enhance the INP content in the sample solution, the volume of filtered nanopure water was reduced from 8 to 5 ml for the samples collected onboard the aircraft. The resulting aerosol suspensions were then analyzed with the INSEKT by pipetting volumes of 50 µL into two 96-well polymerase chain reaction (PCR) plates. Typically, 32 wells were filled with nanopure water while the remaining 160 wells were used to analyze two samples at once. For each sample, 24 wells were filled with the undiluted suspension, 24 wells were filled with a 10 or 15-fold diluted suspension, and 32 wells were filled with a 100 or 225-fold diluted suspension(Schneider et al., 2021).

The INP concentrations reported here have been corrected for the background freezing levels of filtered nanopure water. The INP concentrations extracted from the aircraft samples were further corrected for the INP concentration derived from handling blank filters, which were collected onboard the aircraft without ambient air flowing through the membranes. Then, as the INP concentrations measured from the aircraft filters were rather low and close to the background signal derived from the handling blank filters (Fig. A2a), only the INP concentrations that were at least twice as high as the average background INP concentrations were considered significant and used in this study. More information concerning the handling blank correction can be found in Appendix A1. The INP concentrations extracted from the ground-level samples were well above the INP concentration derived from ground-level handling blank filters (Fig. A2b) and were therefore not corrected further. Finally, the concentration was converted to INP concentration per standard liter of sampled air using standard conditions of 273.15 K and 1013 hPa. In total, from the 19 measurement flights, 18, 16, and 16 filters were collected in the boundary layer, in the free troposphere, and at ground level, respectively.

## 2.2 Aerosol and meteorological measurements at SMEAR II

Comprehensive atmospheric measurements have been ongoing at the SMEAR II station since 1996 (Hari and Kulmala, 2005). The station is surrounded by boreal coniferous forests dominated by Scots pine trees, and the conditions at the site are typical for a background location, with the main pollution sources being the city of Tampere and the activity and buildings at the station.

In this study, we use data from the SMEAR II differential mobility particle sizer (DMPS; Aalto et al., 2001) and aerodynamic particle sizer (APS; TSI model 3321). The DMPS and the APS measure aerosol number size distributions in the size ranges 3-1000 nm in mobility diameters and 0.5-20 µm in aerodynamic diameters, respectively. The data from the DMPS and the APS were combined by converting the aerodynamic diameters measured with the APS to electrical mobility diameters, which are used with the DMPS. To do so, the aerodynamic diameter was divided by the square root of the effective density of the aerosol particles which was estimated to be 1.5 g cm$^{-3}$ from previous studies (Järvi et al., 2009; Kannosto et al., 2008; Khlystov et al.,

2004; McMurry et al., 2002; Stein et al., 1994). More information concerning the operation and sampling conditions of the DMPS and APS at the time of the HyICE-2018 campaign can be found in Brasseur et al. (2022).

We also use global shortwave solar radiation data which was measured above the forest canopy at 67.2 m a.g.l. in the SMEAR II mast using a four-component net radiometer (Kipp & Zonen model CNR4), as well as ambient air temperatures recorded at 4.2 and 67.2 m a.g.l. in the mast using radiation shielded and ventilated platinum-wire thermistors (PT-100), and air pressure measured at ground level (180 m above sea level) using a barometer (Druck DPI 260).

## 2.3 Boundary layer estimation

The atmospheric boundary layer is defined as the lowest part of the troposphere that is directly influenced by the planetary surface and as such is prone to turbulence and strong vertical mixing. Its structure consists of several sub-layers that are formed due to diurnal variations of temperature and heat transfer (Stull, 2017). During spring and summer at SMEAR II, surface-driven convection is the main cause of mixing in the boundary layer during the day (Manninen et al., 2018), and therefore most boundary layers are convective. A schematic diagram of the diurnal evolution of the convective boundary layer over land is presented in Fig. 2. During daytime, a mixed layer is formed via convective turbulence. At the top of the mixed layer, there is a stable layer called the entrainment zone where less turbulent air from above is entrained into the mixed layer below, contributing to the growth of the mixed layer. At times, this stable layer is strong enough to be classified as an inversion (i.e., temperature increases with height). At night, this capping inversion can remain at the top of the residual layer, which contains the pollutants and moisture from the previous mixed layer, even though the turbulence below has weakened. The free troposphere, sometimes called free atmosphere, comprises the air between the top of the boundary layer and the tropopause. In contrast to the boundary layer, the free troposphere is mostly unperturbed by turbulence related to heat transfer.

Boundary layer dynamics directly influence the vertical distribution of atmospheric particles, including INPs. For example, convective mixing occurring in the boundary layer can lift particles originating from near the surface to higher altitudes where they can then be transported to other regions via long-distance transport. Depending on the aging and mixing processes that they undergo in the atmosphere, the physical and chemical properties of the particles, as well as their ice nucleating abilities, can be altered (Després et al., 2012).

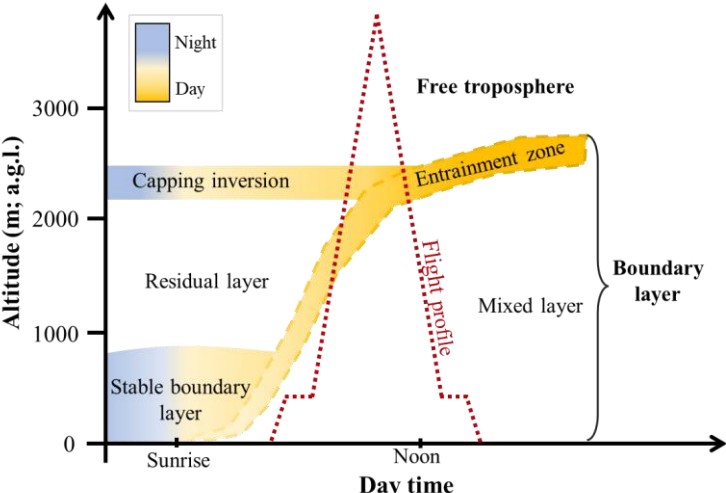

**Figure 2: Schematic diagram of the boundary layer diurnal development adapted from Stull (2017) and Lampilahti et al. (2021), and overlaid with an example flight profile. The actual layer heights may vary from the values depicted on the vertical axis.**

In this study, we use the term boundary layer to represent the layer that encompasses all of the aforementioned lower atmospheric layers (mixed, residual layer, stable boundary layers, capping inversion, and entrainment zone), and we are interested in comparing the INP concentrations measured in the boundary layer to those measured in the free troposphere. As mentioned previously, the boundary-layer depth was estimated subjectively during the flights by monitoring the real-time measurements of particle and water vapor concentration and potential temperature. Indeed, the limit between the boundary layer and free troposphere can usually be identified by a temperature inversion and a drop in the water vapor and particle concentrations (Stull, 2017). After the flights, data from a Halo Photonics Stream Line scanning Doppler lidar located at SMEAR II was used to estimate the limit between the boundary layer and the free troposphere, for comparison with the aircraft measurements. The Halo Doppler lidar was configured with vertically-pointing stare and conical scans (i.e., with velocity azimuth display (VAD) scans at 30 ° elevation angle) repeating every 30 minutes. Additional scans during the 30-minute scan cycle were not used in this study. The range resolution of the lidar is 30 m, with a minimum range of 90 m a.g.l.. More details on the Doppler lidar at SMEAR II can be found in e.g., Hellén et al. (2018). The data was post-processed following Vakkari et al. (2019): horizontal winds were retrieved from the VAD scans and the variance of vertical wind velocity was calculated from 12 consecutive vertical stare measurements. The instrumental noise contribution to the observed variance of vertical wind velocity was estimated from the post-processed signal-to-noise-ratio according to Pearson et al. (2009) and subtracted before calculating turbulent kinetic energy (TKE) dissipation rate profiles according to the method by O'Connor et al. (2010). Finally, the mixed layer height was estimated from the TKE dissipation rate profiles using a threshold of $10^{-4}$ $m^2$ $s^{-3}$, similar to what was done in Hellén et al. (2018). Note that, in some cases, the mixed layer height estimated from the lidar is a lower limit estimate, as the lidar signal can be fully attenuated before first non-turbulent measurements.

Data from the 94 GHz FMCW Doppler cloud radar (RPG-FMCW-94-DP) was used to check the presence of clouds during the flight measurements.

## 2.4 Trajectory models

To identify the origin of the air masses sampled in the free troposphere and investigate potential links between air mass
trajectories and INP concentrations, backward trajectories were calculated with the Hybrid Single-Particle Lagrangian Integrated Trajectory (HYSPLIT) model. The model was used with Global Data Assimilation System (GDAS) meteorological fields (Rolph et al., 2017; Stein et al., 2015) and one 72-hour backward trajectory was computed for each flight, with a release altitude of 3500 m a.g.l. and a starting time corresponding to the time during the flight when the aircraft first reached the free troposphere.

In addition to the HYSPLIT trajectories, we used the Lagrangian FLEXible PARTicle (FLEXPART v10.4) dispersion model to investigate one particular flight where higher INP concentrations were observed in the free troposphere. We ran FLEXPART with increased temporal, horizontal and vertical resolutions compared to the HYSPLIT trajectories to allow for further characterization of this event. FLEXPART was used to calculate the potential emission sensitivity (PES) fields, where PES is the response function of a source-receptor relationship which estimates the potential source contributions for a given receptor
(in this case the measurement site SMEAR II). The PES is therefore proportional to the residence time of the air mass in a specific grid cell, and it was calculated in units of seconds. High values of PES indicate source regions where emissions are likely to significantly impact the tracer concentration at the receptor (Pisso et al., 2019; Seibert and Frank, 2004; Stohl et al., 2005). The simulations were computed for a passive air tracer for which the wet and dry removal processes have no impact. European Center for Medium-Range Weather Forecasts (ECMWF) ERA5 reanalysis meteorology with 137 height levels, 1
295 hour temporal and 0.5° x 0.5° spatial resolution was used as an input to FLEXPART (Hersbach et al., 2018b, a). The air mass history was simulated 3 days backwards in time and arriving at SMEAR II, every hour, with a release at the average altitude of the flight in the free troposphere (3 km a.g.l.). The output resolution was set to 41 height levels spanning from 50 m to 10 km with a vertical resolution of 250 m.

## 3 Results

### 3.1 Campaign overview

The meteorological conditions at SMEAR II during the aircraft measurement campaign are presented in Fig. 3, where the 19 flights are highlighted (grey vertical bands in Fig. 3). A summary of the flight dates and times is available in Table A1.

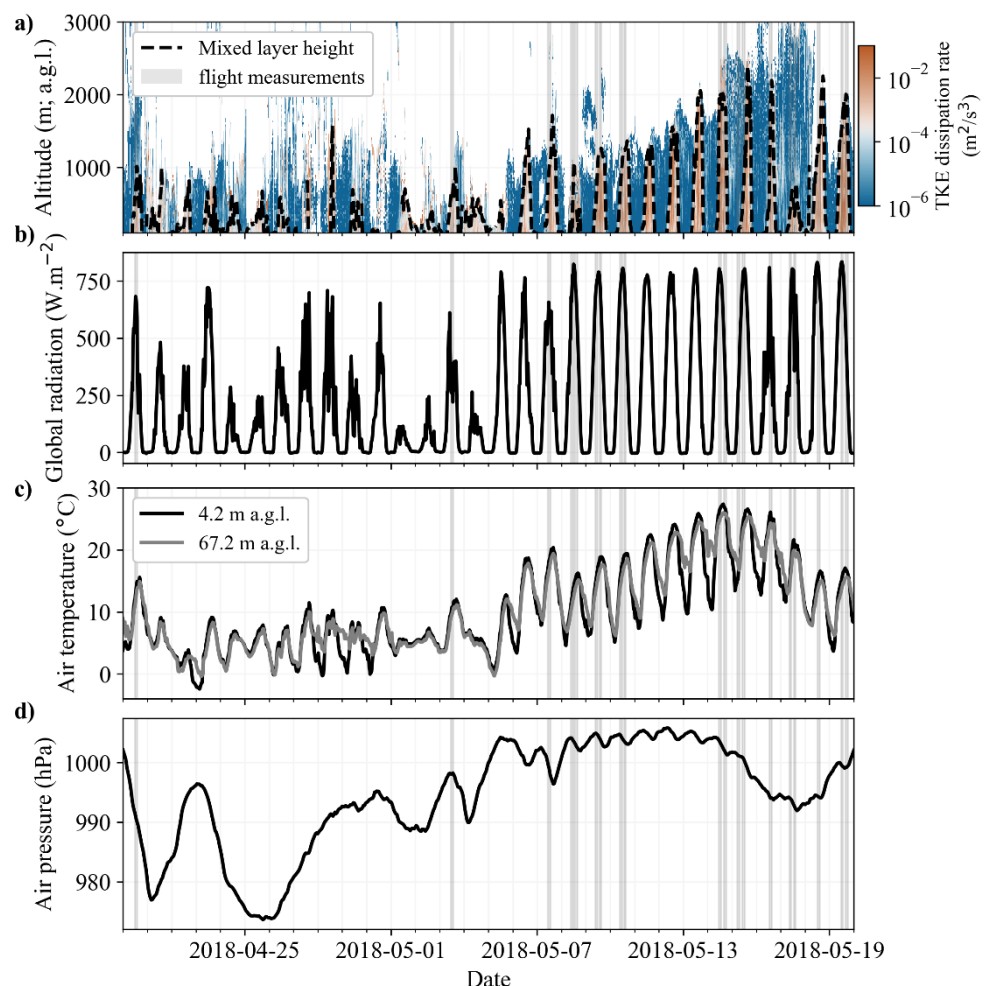

**Figure 3: Overview of a) the mixed layer height, b) global radiation, c) air temperature, and d) air pressure at SMEAR II for the duration of the flight campaign. The mixed layer height was estimated using the TKE dissipation rate from the Doppler lidar. The global shortwave solar radiation was measured from the SMEAR II mast at 67.2 m a.g.l., while the air temperature is shown for measurements at both 4.2 and 67.2 m a.g.l. (at ground level and above the forest canopy). The air pressure was measured at ground level in SMEAR II (180 m above sea level). The flight measurement windows are highlighted with the vertical grey bands.**

There is a clear seasonal change from spring to summer seen in the air temperature measurements (Fig. 3c). During the first period of the flight campaign (from 20 April 2018 to 04 May 2018) ground-level temperatures were relatively cool with an average temperature of 5.3 °C (SD = 3.1 °C) compared to the second period of the flight campaign (from 05 May 2018 to 19 May 2018), when the average ground-level temperature was 14.8 °C (SD = 6.1 °C). Note that May 2018 was exceptionally warm in Finland, and monthly averaged temperature anomalies greater than +4 °C were recorded at several locations (Sinclair et al., 2019). There is also a clear increase in the global shortwave solar radiation during the second period of the campaign (Fig. 3b). Moreover, increased cloud cover in April often disrupts the measured shortwave radiation, while May 2018 had relatively few cloudy days, as illustrated by the clear and consistent sinusoidal diurnal radiation cycle. The seasonal change

also affects the day length with an increase of approximately 2 hours and 45 minutes of daylight between 20 April and 19 May 2018 (Table A1).

Because variations in temperature and heat transfer influence the boundary layer and its diurnal cycle, the seasonal change is also noticeable in the mixed layer height estimated from the SMEAR II lidar measurements (Fig. 3a). There is a rapid increase in the daytime mixed layer height during the second period of the flight campaign, with higher peaks and stronger diurnal cycles than in the first period of the campaign. This agrees well with long-term observations at the SMEAR II station, which show that the deepest boundary layers of all months usually occur in May (Sinclair et al., 2022). In Fig. 3d, the air pressure fluctuates between 975 and 1000 hPa at the beginning of the campaign before increasing to approximately 1004 hPa after 04 May 2018. The second half of the campaign, when most measurements flights were organized, is therefore characterized by relatively warm temperatures, increased solar radiation and air pressure, and deep boundary layers. The relatively high pressures, together with clear skies and high solar radiation, means that winds were low and long-range transport might have been minimal during this part of the campaign.

## 3.2 Vertical distribution of INPs above Hyytiälä

The INP concentrations extracted from the ground-level, boundary-layer and free troposphere samples are shown in Fig. 4a together with the ground-based 24-hour measurements from Schneider et al. (2021) also conducted at SMEAR II. Only the data from Schneider et al. (2021) collected between 20 April and 19 May 2018 is used here in order to cover the same period as the flight campaign. Compared to the ground-level samples presented in this study, the samples used in Schneider et al. (2021) were collected from the aerosol cottage (approximately 20 m from the measurement container) using a $PM_{10}$ inlet with an inlet height of approximately 4.6 m a.g.l. In Fig. 4, the data is presented in the form of boxplots calculated for each activation temperature, where the line dividing the boxes in two represents the median value of the distribution, the lower and upper edges of the boxes represent the 25[th] and 75[th] percentiles, respectively, the lower and upper whiskers represent the minimum and maximum, respectively, and the outliers are represented as single point markers. The raw INP temperature spectra used to produce the box plots can be found in Fig. A2. Note that representing the data in this way might introduce some bias when the number of observations used to calculate the boxplots is more limited, for example at colder temperatures where some of the ground-level INP concentrations appear to be decreasing with decreasing temperature. The number of observations for each sample type as a function of temperature is highlighted in Fig. 4c. The INP concentrations measured at ground level range from $10^{-2}$ to $10^{-1}$ std L$^{-1}$ at the highest temperatures and in the range $10^{0}$-$10^{2}$ std L$^{-1}$ at the lowest temperatures. Overall, these concentrations coincide with the INP concentrations reported by Schneider et al. (2021) for the 24-hour samples collected between 19 April and 20 May 2018, although they have a 3 °C colder ice onset temperature (temperature at which the first ice nucleation event is observed). This is likely due to the shorter sampling time used for the ground-level samples presented here (limited to approximately 3 hours to match the flight duration), which decreased the upper temperature detection limit of INSEKT. The INP concentrations measured in the boundary layer range from about $10^{-2}$ to $10^{0}$ std L$^{-1}$ at the highest temperatures and in the range $10^{1}$-$10^{2}$ std L$^{-1}$ at the lowest temperatures. These concentrations are within the same order of

magnitude as the ground-level and 24-hour measurements from Schneider et al. (2021), although they also have a colder ice onset temperature (approximately 2.5 °C and 5.5 °C colder than the ice onset temperatures of the ground-level and 24-hour measurements, respectively) likely due to shorter sampling times as well ($\approx$ 70 minutes for the boundary-layer samples). On the other hand, the INP concentrations measured in the free troposphere range from $10^{-2}$ to $10^{-1}$ std L$^{-1}$ at the highest temperatures and from $10^{-1}$ to $10^{1}$ std L$^{-1}$ at the lowest temperatures, and they are significantly lower than the INP

concentrations measured in the boundary layer and at ground level. They also have an ice onset temperature colder than any other measurements shown in this study (approximately 4.5°C, 7°C and 10°C colder than the ice onset temperatures of the boundary-layer, ground-level and 24-hour measurements from Schneider et al. (2021), respectively). As mentioned previously, this is likely due to shorter sampling times used for the free troposphere samples ($\approx$ 60 minutes).

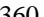

**Figure 4: (a) INP temperature spectra and (b) activated fraction as a function of the activation temperature for all samples collected during the aircraft measurement campaign together with the ground-level data from Schneider et al. (2021) collected in Hyytiälä from 20 April to 19 May 2018. The activated fraction was calculated as the ratio between the INP concentration and the number**

concentration of particles larger than 300 nm using the data from the OPS and the combined DMPS-APS for the aircraft and the ground-level samples, respectively. In a) and b), the point markers indicate outlier observations. (c) Number of observations for each sample type as a function of temperature.

In Fig. 4b, the activated fraction, calculated as the ratio of the INP concentration to the number concentration of particles larger than 300 nm, is presented. This size range was selected based on previous studies showing a relationship between INP concentration and aerosol number concentration for particles larger than 300 nm in diameter (e.g., DeMott et al., 2003b; Richardson et al., 2007). Figure 4b shows that there is more overlap between the activated fraction from all sample types compared to the INP concentrations shown in Fig. 4a. The activated fraction from the ground-level and boundary-layer samples are within the same order of magnitude, while the activated fraction from the free troposphere samples is overall lower, with some overlap with the ground-level samples below -20 °C. This suggests that, even though particles sampled in the free troposphere are overall less efficient INPs, there are a few cases where the free tropospheric INPs are as efficient as those sampled at ground level. These specific cases are further discussed in section 3.7. The ice nucleation active surface site (INAS) densities, calculated as the ratio of the INP concentration to the surface area concentration of particles larger than 300 nm, are presented in Fig. A3.

### 3.3 Particle concentrations and size distributions above Hyytiälä

The median particle concentrations and size distributions measured at ground level, in the boundary layer and in the free troposphere calculated from the 19 flights are shown in Fig. 5. The submicron size distribution measured at ground level (green data points in Fig. 5a) exhibits the characteristic modal structure found at the SMEAR II station (Dal Maso et al., 2005), with a nucleation mode observed in the size range of 3-25 nm, and an Aitken mode (25-100 nm) growing into an accumulation mode (100-500 nm). The size distribution measured aloft in the boundary layer (blue data points in Fig. 5a) shows very similar features. The lack of observed sub-10 nm nucleation mode in the boundary layer is likely due to the higher cut-off size of the aircraft SMPS (10 nm) compared to the ground-level DMPS (3 nm). In addition, very low concentrations of coarse mode particles above 1000 nm are measured both at ground level and in the boundary layer. These results agree with previous measurements conducted at SMEAR II, which show that the aerosol size distribution measured at 300 m a.g.l. compared well to ground-level observations (Schobesberger et al., 2013). In addition, the concentration of particles > 300 nm measured in the boundary layer (median of $\approx 26.6$ cm$^{-3}$) is very similar to the concentration measured at ground level (median of $\approx 28.8$ cm$^{-3}$), as seen in Fig. 5b, and agrees well with previous aircraft measurements conducted at SMEAR II over relatively similar size ranges (Väänänen et al., 2016).

The free troposphere is characterized by a much lower concentration of particles > 300 nm (median of $\approx 10.0$ cm$^{-3}$) compared to the ground-level and boundary-layer observations (Fig. 5b). This agrees well with previous aircraft measurements conducted above SMEAR II, which also indicate a sharp decrease in particle concentration when the free troposphere is reached (Beck et al., 2022; Lampilahti et al., 2021; Schobesberger et al., 2013; Väänänen et al., 2016). Furthermore, the particles in the free troposphere have a very different particle number size distribution pattern (orange data points in Fig. 5a). There are no apparent

nucleation mode particles below 25 nm, and the Aitken mode growing into the accumulation mode has much lower concentrations than observed at ground level and in the boundary layer. In addition, the particle concentration in the coarse mode (> 1000 nm) is systematically lower than what is observed in the boundary layer.

Note that, in Fig. 5a, we observe that higher concentrations of particles > 2000 nm are measured with the OPS in the boundary layer and free troposphere compared to ground-level measurements conducted with the APS. This deviation, which was observed for each flight measurements (e.g., also in Fig. 9j-l), is likely due to instrumental differences and has been observed in a previous laboratory study where the OPS and APS were compared (Zerrath et al., 2011). As explained in Zerrath et al. (2011), particle sizing deviates between the OPS, which uses optical diameters, and the APS, which uses aerodynamic diameter (which was converted to mobility diameter here when combining the APS and DMPS data shown in Fig. 5). Such deviation is especially true for diameters > 1000 nm where the refractive index of the aerosol can significantly affect the intensity of light scattered (Szymanski et al., 2009) and detected by the optical sizer. Although Mie correction can be applied to size distributions of known particles, correcting ambient aerosol data is not trivial and thus we do not explore it further in this study.

The similarities between the size distributions and particle concentrations measured at ground level and in the boundary layer, together with the similar INP concentrations and activated fractions, suggest a similar aerosol population is sampled between the surface and the boundary layer aloft. In other words, it appears that the boundary layer was well-mixed during the aircraft measurements and that particles from the surface were efficiently transported and mixed within the boundary layer, which is consistent with the TKE dissipation rate profiles from the SMEAR II Doppler lidar (Fig. 3a). Thus, we hypothesize that the INPs encountered in the boundary layer above the boreal forest are dominated by local and regional sources at the surface, at least during the spring/summer season. Moreover, because the INP concentrations and activated fractions measured at ground level and in the boundary layer are similar to those reported by Schneider et al. (2021) for the same time period, it is possible that similar INPs were sampled in both studies, which Schneider et al. (2021) relate to local biogenic particles rather than long-range transported particles. It is therefore possible that biogenic particles represent an important fraction of the INPs sampled at ground level and in the boundary layer in this study. A recent study from Maki et al. (2023) showed that airborne microorganisms from forested areas could maintain similar concentrations from ground level up to 500 m under efficient vertical mixing conditions. Since the boundary layer was well-mixed during the aircraft campaign, we can expect that surface biogenic particles had a non-negligible impact on the INPs sampled in the boundary layer. However, such a hypothesis cannot be confirmed with the data presented in this work, and more measurements, such as heat treatment tests (e.g., Hill et al., 2016), would be needed to examine the presence of biogenic INPs in the samples."

On the other hand, the lower INP concentrations measured in the free troposphere are most likely due to the lower particle concentrations encountered there combined with the fact that the free tropospheric particles might be less efficient INPs, as suggested by the overall lower activated fractions (Fig. 4b). In addition, the differences observed in the size distribution pattern suggest that the aerosol populations present in the free troposphere are different than those encountered in the boundary layer and at ground level. It is likely that these particles, and thus the INPs, were transported from distant regions via long-range transport, as discussed in section 3.4. Nevertheless, as mentioned previously, the activated fraction from the free troposphere

samples sometimes overlap with the rest of the observations, in particular with the ground-level measurements from Schneider et al. (2021) and at temperatures below -20 °C. This shows that there are some cases where the activated fraction of the free troposphere samples is higher and similar to those observed at ground level. Such observation raises the question of whether

surface particles might influence the free troposphere locally in some way. This question is further investigated in section 3.7, where we focus on the flights with the highest INP concentrations observed in the free troposphere.

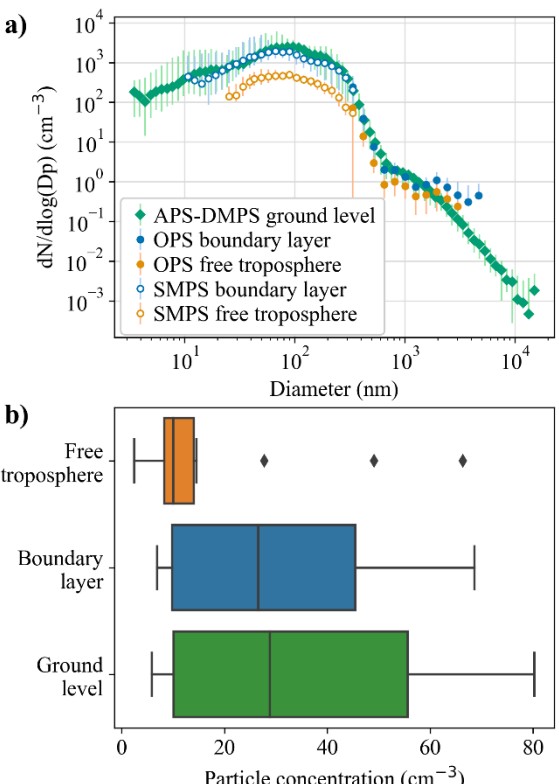

**Figure 5: a) Median particle number size distributions calculated from ground-level measurements (SMEAR II APS and DMPS) as well as boundary layer and free troposphere measurements (aircraft SMPS and OPS) over the 19 flights of the campaign. The error bars represent the 25th and 75th percentiles. The size distribution shown with a linear scale can be seen in Fig. A4. b) Box plots of the concentration of particles > 300 nm measured at ground level, in the boundary layer and in the free troposphere calculated over all the flights.**

### 3.4 Origin of the air masses in the free troposphere

In Fig. 6a, we show the HYSPLIT 72-hour backward trajectories of the air masses arriving at 3500 m a.g.l. in the free troposphere at the time when the aircraft reached the free troposphere, together with the altitude of the trajectories over time (Fig. 6b) and the INP temperature spectra of the corresponding free troposphere samples (Fig. 6c).

Most of the free tropospheric air masses originate from the west and remain above 3500 m a.g.l. for the duration of the

450 simulation. Two groups of air masses can be differentiated based on their trajectories. The first group of air mass trajectories, corresponding to the first and last days of the measurement period in May 2018 shown in Fig. 6, are longer and cover large distances (> 3000 km), some coming from as far as the Hudson Bay in northeastern Canada (light green trajectory from 8 May 2018 in Fig. 6a). Some of these air masses cross the North Atlantic Ocean before reaching Northern Europe and are therefore mostly maritime (e.g., green and purple trajectories from 8 and 18 May 2018 in Fig. 6a), while others cover slightly shorter

distances and travel over both continents and seas (e.g., dark blue and brown lines from 3 and 19 May 2018 in Fig. 6a, respectively). This group of longer air mass trajectories have very similar INP concentrations, which correspond to the lowest concentrations in the INP temperature spectra presented in Fig. 6c. The fact the INP concentrations vary over a narrow range (less than one order of magnitude) makes it difficult to identify possible links between air mass trajectory and INP concentrations for this specific group of air masses.

The second group of air mass trajectories, between 10 and 16 May 2018, are shorter (<1000 km) and more regional, circulating mostly over Northeastern Europe. Some of these trajectories have clear anticyclonic paths (e.g., oranges lines from 14 and 15 May 2018 in Fig. 6a). These observations coincide with the high pressures observed at the same time over SMEAR II (Fig. 3d), where long-range transport is expected to be minimal. Most of the INP concentrations corresponding to these air masses fall in the same range of low concentrations as the longer trajectories discussed above. However, two of the air masses, on 15

and 16 May 2018, correspond to the highest INP concentrations measured in the free troposphere during the flight campaign. These specific measurements will be further discussed in section 3.7.

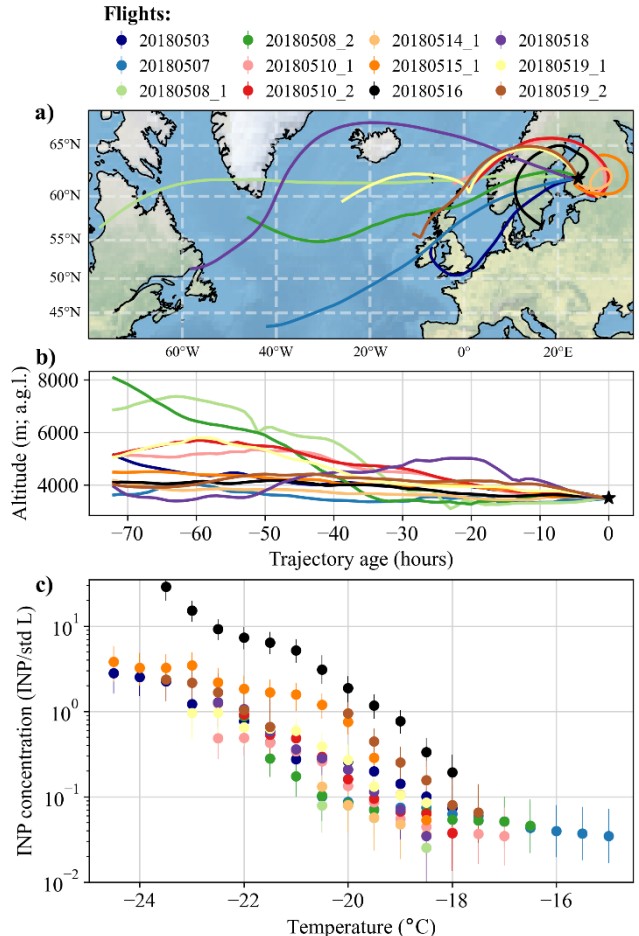

**Flights:**
- 20180503
- 20180507
- 20180508_1
- 20180508_2
- 20180510_1
- 20180510_2
- 20180514_1
- 20180515_1
- 20180516
- 20180518
- 20180519_1
- 20180519_2

**Figure 6. a) HYSPLIT 72-hour backward trajectories and b) altitude of the trajectories over time for the air masses arriving at 3500 m a.g.l. in the free troposphere above SMEAR II at the time when the measurement aircraft reached the free troposphere. c) INP temperature spectra of the free troposphere samples corresponding to each flight. In each plot, the color represents a specific flight, as indicated in the legend above panel a). Note that there were sometimes two flights per day, and each flight is identified by a number (_1 or _2) in the legend above panel a). In a) and b), the black star represents the measurement location in the free troposphere above SMEAR II, at 61°51′ N, 24°17′ E and 3500 m a.g.l.**

### 3.5 Comparison to existing parameterizations

In Fig. 7a-f, the INP concentrations measured in the boundary layer and in the free troposphere are compared to INP concentrations predicted by three existing parameterizations from Schneider et al. (2021), DeMott et al. (2010), and Tobo et al. (2013), which are presented in Table 1. This section focuses on the aircraft INP measurements conducted at altitude, and a detailed comparison between parameterizations and INP concentrations from ground-based filter measurements similar to those presented here can be found in Schneider et al. (2021). Schneider et al. (2021) used 15 months of measurements in Hyytiälä (from March 2018 to May 2019 with a time resolution between 24 and 144 hours) to investigate the seasonal cycle of INP concentration in the Finnish boreal forest and concluded that variations were driven by the abundance of biogenic

aerosols emitted from vegetation in the forest. They developed a new non-aerosol-specific parameterization using ground-level ambient air temperature as a proxy for the seasonal change. For the boundary-layer samples (Fig. 7a), the Schneider et al. (2021) parameterization is calculated using the ground-level ambient air temperature measured at 4.2 m a.g.l., while for the free troposphere samples (Fig. 7b), the parameterization is calculated using the ambient air temperature measured onboard the aircraft in the free troposphere. In both cases, the ambient temperature was averaged over the sampling time of each sample. The parameterization by DeMott et al. (2010) was developed by combining observations from nine different field studies (in Colorado, eastern Canada, the Amazon, Alaska, and the Pacific basin) collected via aircraft measurements using a continuous flow diffusion chamber (CFDC). It is considered as a global aerosol type-independent parameterization for atmospheric particles of nonspecific composition and uses the total number concentration of particles with diameters larger than 0.5 µm. Tobo et al. (2013) proposed a modified version of the parameterization from DeMott et al. (2010) using observations from a ponderosa pine forest in Colorado. To calculate the total number concentration of particles with diameters larger than 0.5 µm used in these two parameterizations, we used the SMEAR II APS data for the boundary-layer samples (Fig. 7c, and e). This choice was motivated by the similarities between the size distributions and particle concentrations measured at ground level and in the boundary layer, the fact that the boundary layer was well-mixed during the aircraft measurements, and to investigate if ground-level measurements can be used in parameterizations to predict INP concentrations observed aloft in the boundary layer. For comparison, the INP concentrations predicted by the DeMott et al. (2010) and the Tobo et al. (2013) parameterizations calculated using the aircraft OPS data is shown in Fig. A5. On the other hand, since the free troposphere was characterized by distinct size distributions and particle concentrations, we used the aircraft OPS data to calculate the total number concentration of particles with diameters larger than 0.5 µm in the free troposphere (Fig. 7d, f, and g). For both sample types, the particle concentration was averaged over the sampling time of each sample.

**Table 1: Overview of the INP parameterizations used in this study together with the temperature range for which they have been developed and the input parameters used in each parameterization.**

| Reference | Temperature range | Equation | Input parameters |
|---|---|---|---|
| Schneider et al. (2021) | -25 to -12 °C | $n_{INP} = 0.1 \cdot exp(a1 \cdot T_{amb} + a2) \cdot exp(b1 \cdot T + b2)$ <br> with $a1 = 0.074$ K$^{-1}$, $a2 = -18$, $b1 = -0.504$ K$^{-1}$, and $b2=127$ | Ground-level ambient air temperature $T_{amb}$ (K) <br> Activation temperature $T$ (K) |
| DeMott et al. (2010) | -35 to -9 °C | $n_{INP} = a(273.16 - T)^b (n_{AP,>0.5\,\mu m})^{(c(273.16-T)+d)}$ <br> with $a = 0.0000594$, $b = 3.33$, $c = 0.0264$, and $d = 0.0033$ | Number concentration of particles with diameters larger than 0.5 µm $n_{AP,>0.5\,\mu m}$ (cm$^{-3}$) <br> Activation temperature $T$ (K) |
| Tobo et al. (2013) | -34 to -9 °C | $n_{INP} = exp(\gamma(273.16 - T) + \delta)(n_{AP,>0.5\,\mu m})^{(\alpha(273.16-T)+\beta)}$ <br> with $\gamma = 0.414$, $\delta = -6.671$, $\alpha = -0.074$, and $\beta = 3.8$ | |
| Adjusted Tobo et al. (2013) | -34 to -9 °C | $n_{INP} = exp(\gamma(273.16 - T) + \delta)(n_{AP,>0.5\,\mu m})^{(\alpha(273.16-T)+\beta)}$ <br> with $\gamma = 0.7408$, $\delta = -16.0788$, $\alpha = 0.2746$, and $\beta = -3.3184$ | |

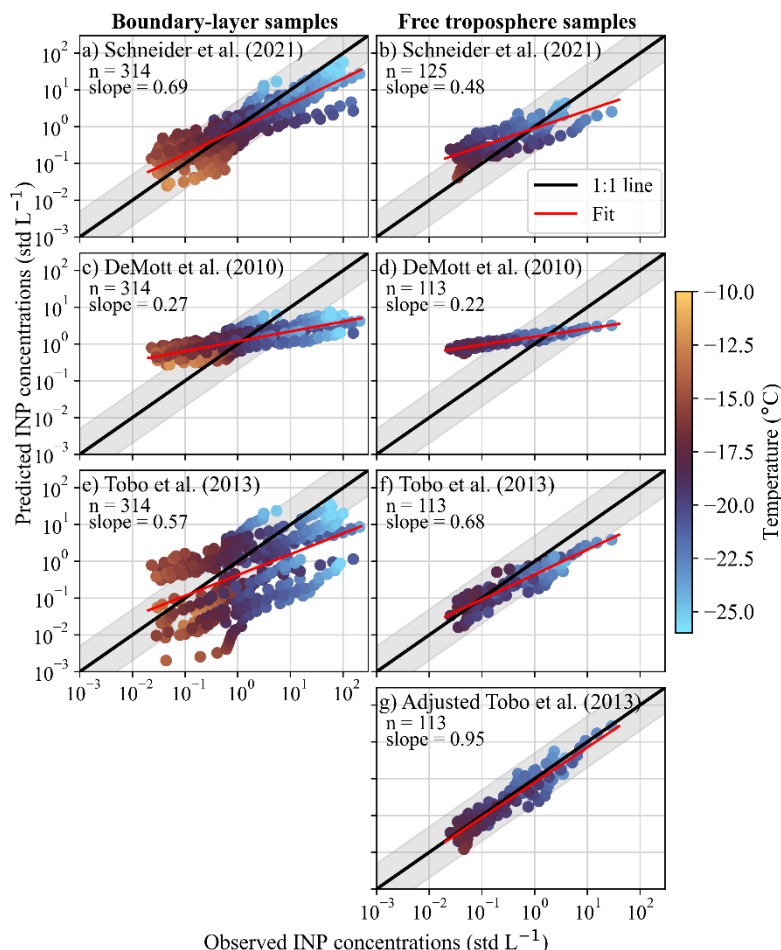

**Figure 7.** Comparison between the INP concentrations observed in the boundary layer (left side) and the free troposphere (right side) and the INP concentrations predicted using the parameterizations from a) and b) Schneider et al. (2021), c) and d) DeMott et al. (2010), and e) and f) Tobo et al. (2013). The black solid line represents the 1:1 line while the grey shaded area indicates a range of a factor of 5 from the 1:1 line. The red solid lines show a linear regression fit through the logarithmically transformed data points. The slope of the fit and the number of data points used is shown in each panel.

Among all the investigated parameterizations, Schneider et al. (2021) reproduce most of the boundary layer data points by predicting 90 % and 44 % of the measurements within a factor 5 and 2, respectively. Therefore, even though it was developed to represent the concentration of INPs in Finnish boreal forests near the surface, the Schneider et al. (2021) parameterization also reproduces INP concentrations in the boundary layer over the same environment. Our aircraft measurements are however limited to the spring/summer season, and additional measurements conducted at different times of year would be necessary to determine the ability of the parameterization to predict INP concentrations in the boundary layer above Hyytiälä. DeMott et al. (2010) reproduce 62 % and 31 % of the data points within a factor of 5 and 2, respectively, but the slope of its linear regression fit is more shallow, and it does not match the temperature trend. It overestimates the INP concentrations measured at temperatures warmer than about -18 °C where the concentrations are less than $\approx 1$ L$^{-1}$, likely due to the fact that the

parameterization was based on CFDC measurements without using an aerosol concentrator where high uncertainty is expected for the detection of low INP concentrations, as discussed in Tobo et al. (2013). On the other hand, the DeMott et al. (2010) parameterization underestimates the INP concentrations greater than $\approx 1$ L$^{-1}$, which suggest some differences in the INP population sampled during our study compared to the samples studied in DeMott et al. (2010). Lastly, the parameterization from Tobo et al. (2013) only reproduces 47 % and 16 % of the data points within a factor of 5 and 2, respectively, and underestimates a large part of the INP concentrations measured in the boundary layer. Very similar results are obtained when calculating the DeMott et al. (2010) and Tobo et al. (2013) parameterizations using the aircraft OPS data instead of the SMEAR II APS data (Fig. A5), where DeMott et al. (2010) reproduces 60 % and 32 % of the data points within a factor of 5 and 2, respectively, while Tobo et al. (2013) reproduces 49 % and 17 % of the data points within a factor of 5 and 2, respectively. This highlights that both ground-level measurements and aircraft measurements from the boundary layer produce similar parameterization results, which suggests that ground-level measurements are sufficient for predicting INP concentrations aloft in the boundary layer. Based on these results, we conclude that, among the parameterizations tested here, the Schneider et al. (2021) parameterization performs best at predicting the concentration of INPs in the boundary layer above a Finnish boreal forest environment. This further supports our hypothesis that the INPs measured in the boundary layer could be local biogenic particles rather than long-range transported particles.

Concerning the free troposphere samples, Schneider et al. (2021) reproduce 77 % and 44 % of the measurements within a factor of 5 and 2, respectively (Fig. 7b). It overestimates most of the INP concentrations measured, which is not necessarily surprising considering that the parameterization is based on near-surface observations. Thus, the Schneider et al. (2021) parameterization performs relatively better at representing the well-mixed boundary layer rather than the more remote free troposphere where INPs can be scarce and originate from distant sources.

On the other hand, the DeMott et al. (2010) parameterization is considered to be suitable for representing a mixture of continental aerosol such as anthropogenic haze, biomass burning smoke, biological particles, soil and road dust (Mamouri and Ansmann, 2016), and one could therefore expect that it would successfully predict INP concentrations observed in the free troposphere. However, DeMott et al. (2010) only reproduce 55 % and 28 % of the measurements within a factor of 5 and 2, respectively. As observed previously, the parameterization overestimates the INP concentrations lower than $\approx 1$ L$^{-1}$ and underestimates the concentrations greater than $\approx 1$ L$^{-1}$.

The Tobo et al. (2013) parameterization reproduces 93 % and 59 % of the measurements within a factor of 5 and 2, respectively. Even though it tends to underestimate the INP concentrations, especially for the colder temperatures (Fig. 7f), it is the parameterization that performs best at predicting the INP concentrations measured in the free troposphere. This is somewhat surprising since Tobo et al. (2013) is considered to be a composition-specific INP parameterization, while our results suggest that the free tropospheric particles and INPs are likely long-range transported and therefore are likely a mixture of various particles. Despite this, it seems that the number concentration of particles with diameters larger than 0.5 µm and temperature dependence described in the Tobo et al. (2013) parameterization successfully represents the free troposphere measurements presented in this study. It is possible that the equation form used in Tobo et al. (2013), which differs slightly from the one used

in DeMott et al. (2010) due to the exponential dependence on temperature of the first term (Table 1), is a better fit for the free troposphere data presented here. To test this hypothesis, we adjusted the coefficients used in the parameterization from Tobo et al. (2013) to better fit our free troposphere data while keeping the same mathematical form. This was done using the in situ observations of number concentration of particles with diameters larger than 0.5 µm and INP concentrations measured in the free troposphere, and following the method described in the supporting information of DeMott et al. (2010). Each fitting was

calculated using the Levenberg-Marquardt algorithm, and the following adjusted coefficients were obtained: $\gamma = 0.7841$, $\delta = -16.9941$, $\alpha = 0.3187$, and $\beta = -4.1788$. As shown in Fig. 7g, the adjusted parameterization reproduces 100 % and 85 % of the data points within a factor of 5 and 2, respectively, and therefore successfully represents the free troposphere INP measurements. This parameterization with the adjusted coefficients should however be used with caution as the number of observations is very limited, and more measurements conducted in the free troposphere would be necessary to efficiently

represent the variations of INP concentrations. Moreover, the fact that none of the pre-established parameterizations presented here perfectly represent the trend in the INP concentrations measured in the free troposphere further stresses the need for additional measurements and characterization of the free tropospheric INPs above the Finnish boreal forest to properly predict the INP concentrations encountered there.

### 3.6 Comparison to previous studies

INP concentrations vary significantly across the world depending on, among other things, the location, time of the year and altitude of the measurements (Kanji et al., 2017). In Fig. 8, we compare our data to literature data collected mostly from aircrafts in different environments.

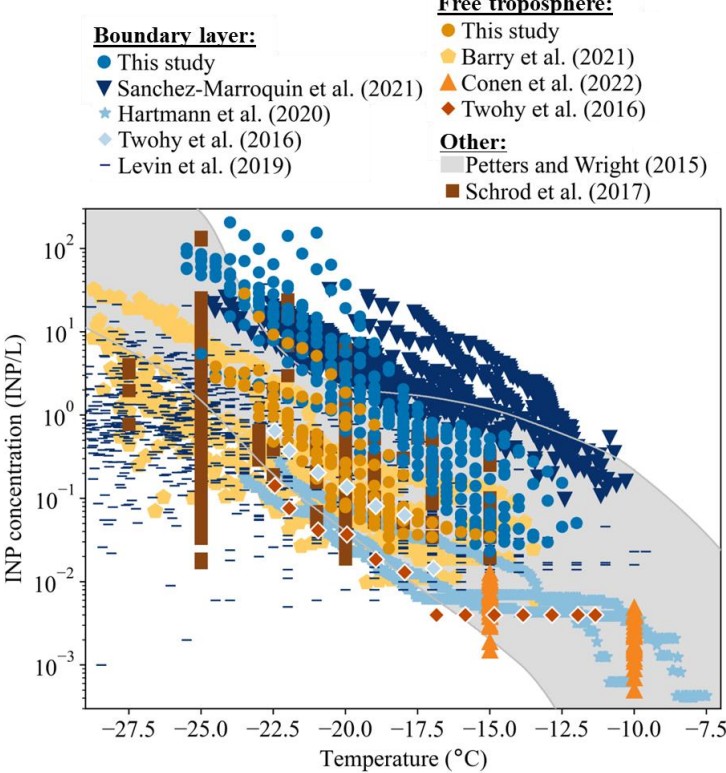

**Figure 8. INP concentrations from the present study compared with literature data. The measurements from Sanchez-Marroquin et al. (2021) were conducted in the boundary layer in the South East of the British Isles, while Hartmann et al. (2020) conducted their measurements in the High Arctic boundary layer. For the Twohy et al. (2016) data collected in the western United States, the light blue diamonds represent a filter sampled within the boundary layer (at 1067 m a.g.l.), while the dark orange diamonds represent a filter sampled primarily in the free troposphere (between 897 and 3638 m a.g.l.). For the Levin et al. (2019) study conducted in California, the data represented here correspond to the measurements made in the boundary layer (below 2 km). The measurements from Barry et al. (2021) were conducted between 1300 and 5100 m above sea level in the western United States. The measurements from Conen et al. (2022) were conducted under free troposphere conditions at Jungfraujoch (3580 m above sea level) in the Swiss Alps. The grey band represents the data range given in Petters and Wright (2015) derived from precipitation samples collected around the world. The measurements from Schrod et al. (2017) were conducted between 500 and 2500 m a.g.l. (likely both in the boundary layer and in the free troposphere) over the Eastern Mediterranean.**

Most of the data presented in this study fall within the mid-latitude data range given by Petters and Wright (2015; grey band in Fig. 8) derived from precipitation samples collected around the world, except for the highest INP concentrations measured between -18 and -24 °C in the boundary layer. Part of the data presented in this study also overlap with some of the INP concentrations reported by Schrod et al. (2017) who sampled Saharan dust plumes over the Eastern Mediterranean.

Concerning the boundary layer measurements, most of the INP concentrations presented in this study are higher than concentrations measured in the marine boundary layer in the Arctic during winter (Hartmann et al., 2020), in coastal California during wintertime (Levin et al., 2019), and above a forested site in the western United States (Twohy et al., 2016). Compared to INP measurements conducted in the South East of the British Isles (Sanchez-Marroquin et al. 2021), the INP concentrations

we observed in the boundary layer are about one order of magnitude lower for temperatures above approximately -18 °C, but are within the same order of magnitude for temperatures below approximately -18 °C.

The majority of the INP concentrations measured in the free troposphere in this study fall within the higher range of free tropospheric measurements from Barry et al. (2021) conducted during wildfire events in western US. The INP concentrations from Conen et al. (2022) observed under free troposphere conditions at Jungfraujoch in the Swiss Alps are relatively lower than the concentrations reported here at similar temperatures (≈ -15 °C). In addition, INP concentrations measured in the free troposphere over a forested site in the western United States (Twohy et al., 2016) are about one order of magnitude lower than

the average INP concentrations observed in the free troposphere in the present study.

Thus, the INP concentrations measured in the boundary layer and in the free troposphere are mostly higher or within the same range as previous measurements from various regions. These observations illustrate that both the boundary layer and the free troposphere above the Finnish boreal forest are relatively rich in INPs with concentrations comparable to other environments.

### 3.7 Case study: higher concentrations of INPs in the free troposphere

During specific flights from 16 afternoon and 17 morning May 2018, INP concentrations measured in the free troposphere were higher than usually reported during the flight campaign. These two flights are compared to the early morning flight of 15 May 2018 which is chosen to illustrate a measurement flight with a more typical vertical distribution of INP concentrations (cf., Fig. 4a). In Fig. 9a-c, the INP temperature spectra of these three consecutive flight days are shown. As mentioned previously, the early morning flight on 15 May 2018 (Fig. 9a) is characterized by similar INP concentrations between ground

level and boundary layer and lower INP concentrations in the free troposphere, as were most flights during this campaign. Conversely, the afternoon flight on 16 May 2018 shows relatively high INP concentrations in the free troposphere, which are within the same order of magnitude as the INP concentrations measured at ground level and in the boundary layer (Fig. 9b). Likewise, the next flight, during the morning of 17 May 2018, also shows higher INP concentrations in the free troposphere. Note that the discontinuity observed in the free troposphere sample from 17 May 2018 occurs at the dilution step. Possible

explanation for this include the inactivation of INPs during dilution, the comparably low amount of sampled aerosol due to the shorter sampling time used for this filter (≈ 45 minutes), or insufficient redispersion of the suspension and the resulting inhomogeneity caused by particle settling (Harrison et al., 2018).

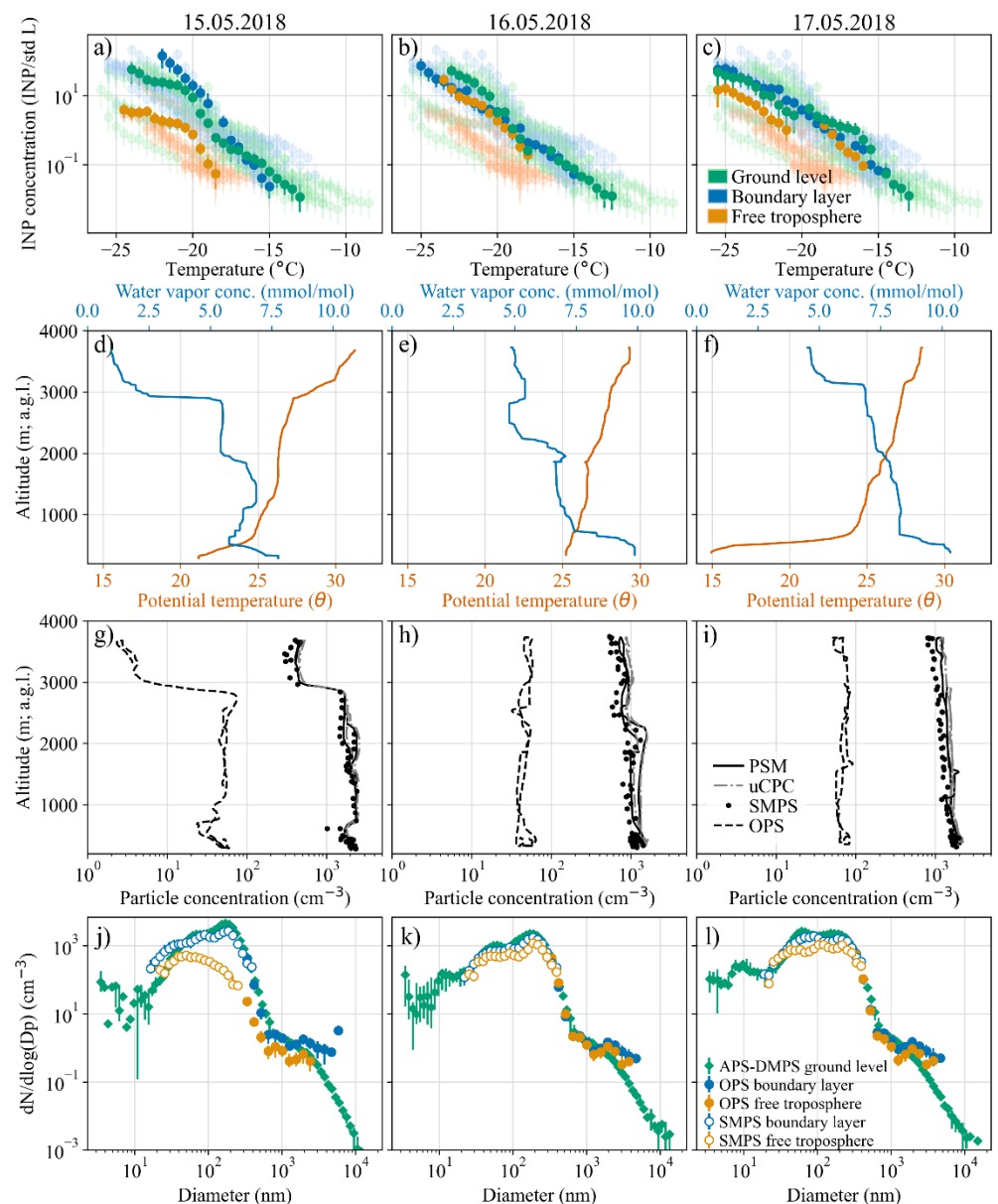

**Figure 9. a), b) and c)** INP temperature spectra for the three flights of the case study (15, 16 and 17 May 2018), plotted on top of the complete dataset (transparent data points) as in Fig. A2c. **d), e) and f)** Potential temperature and water vapor concentration plotted as a function of altitude during the ascents, which occurred between 05:30 and 06:20 (UTC+2) on 15 May; between 13:40 and 14:20 (UTC+2) on 16 May; and between 08:40 and 09:20 (UTC+2) on 17 May 2018. **g), h) and i)** Particle concentrations as a function of altitude. **j), k) and l)** Median particle number size distributions for the three consecutive flights of the case study. The size ranges of the particle counters used onboard the aircraft are > 1.5 nm for the PSM, > 3 nm for the uCPC, > 10 nm for the SMPS and > 300 nm for the OPS.

To better understand the differences between these three days, we examine profiles of meteorological and particle variables measured during the flights. For the flight on 15 May 2018, which took place very early in the morning (starting around 05:00 UTC+2), Fig. 9d shows that there is a sharp decrease in the water vapor concentration and an increase in the potential

temperature at approximately 2800 m a.g.l., indicating the transition between the residual layer and the free troposphere. This
agrees relatively well with the lidar data (Fig. 10a), which shows a residual layer up to approximately 2600 m a.g.l., above a
very shallow mixed layer (under 200 m a.g.l.) which had just started developing and was not sampled at the time of the flight,
and therefore is not visible in (Fig. 9d). The limit between the boundary layer and the free troposphere is also clearly visible
from the measurements of particle concentrations with a sudden decrease in the concentration around 2800 m a.g.l. (Fig. 9g),
which could explain the lower INP concentrations measured in the free troposphere. On the afternoon flight of 16 May 2018,
however, it is difficult to estimate the limit between the boundary layer and the free troposphere using the aircraft
measurements. Indeed, the particle concentration remains relatively high ($\approx$ 40 cm$^{-3}$ for particles > 300 nm) and homogeneous
from 300 to 3500 m a.g.l. (Fig. 9h). Only a small decrease in the particle concentration and water vapor concentration (Fig.
9e), observed at approximately 2400 m a.g.l., hints at a change of atmospheric layer. This is confirmed when looking at the
SMEAR II lidar data presented in Fig. 10b, which also shows a limit between the boundary layer and the free troposphere
between 2000 and 2400 m a.g.l. during the flight window. Hence the higher INP concentrations measured in the free
troposphere on 16 May 2018 are likely due to the high particle concentrations encountered there. Similarly, on the morning
flight of 17 May 2018, the particle concentration is also high ($\approx$ 70 cm$^{-3}$ for particles > 300 nm) and homogeneous between
300 and 3500 m a.g.l, as shown in Fig. 9i. The real-time measurements of potential temperature and water vapor concentration
(Fig. 9f) show a low mixed layer at approximately 500 m a.g.l. and a deep residual layer up to approximately 3000 m a.g.l.,
which are also visible in the lidar data (Fig. 10c). As for 16 May 2018, it seems like the higher INP concentrations observed
in the free troposphere are due to the high particle concentrations present in the free troposphere. It is however unclear from
where these particles and INPs originate. One possibility is that the particles and INPs have been transported from remote
sources to the free troposphere above SMEAR II by long-range transport. Another possibility is that the particles and INPs
have been ventilated out of the boundary layer to the free troposphere locally.
To further investigate the source(s) of these high particle concentrations (and thus INPs) encountered in the free troposphere,
we first compare the particle number size distribution measured between ground level, the boundary layer, and the free
troposphere for each flight (Fig. 9j-l). On 15 May 2018, there is a clear difference between the size distribution measured in
the boundary layer and in the free troposphere (Fig. 9j). This is similar to what was reported for most flights in the study (Fig.
a) and suggests that two distinct aerosol populations were sampled between the boundary layer and the free troposphere.
However, the size distribution measured in the free troposphere on 16 May 2018 is very similar to those measured at ground
level and in the boundary layer on the same day (Fig. 9k). All three median size distributions show similar features and
concentrations; they have a clear Aitken mode around 40 nm, an accumulation mode around 200 nm and rather low
concentrations of coarse mode particles above 1000 nm. This implies that a single aerosol population was sampled from ground
level to the free troposphere. On 17 May 2018, the size distribution measured in the free troposphere is still relatively similar
to those observed at ground level and in the boundary layer. There is however a small deviation in the free troposphere size
distribution between 30 and 90 nm, where the concentration decreases compared to what is observed at ground level and in
the boundary layer. This depletion of particles could be due to cloud processing, which agrees well with the presence of a

cloud between approximately 3000 and 4500 m a.g.l. during the flight on 17 May 2018, as seen from the SMEAR II Doppler cloud radar data (Fig. 10f). Note that because the limit between the boundary layer and the free troposphere was difficult to estimate during the flight of 17 May 2018, it is possible that part of the free troposphere sample was sampled in the residual layer (Fig. 10c). This situation makes it challenging to understand if the relatively high INP concentrations measured in this sample are related to the higher particle concentration encountered in the free troposphere or to a 'contamination' from the residual layer. However, this should matter little if one single aerosol population is present from ground level to the free troposphere, as suggested by Fig. 9l. Moreover, when comparing the size distributions measured in the free troposphere and in the first meters of the residual layer (2000 m a.g.l.), rather similar distributions are observed, especially for particle diameters above 100 nm (Fig. A6). Furthermore, it is important to stress that, although the analysis of the particle number size distributions and concentrations gives valuable information on the vertical distribution and physical mixing state of the aerosol population, such information cannot necessarily be directly extended to the INPs, which represents a very small and highly variable fraction of the overall aerosol population (DeMott et al., 2010).

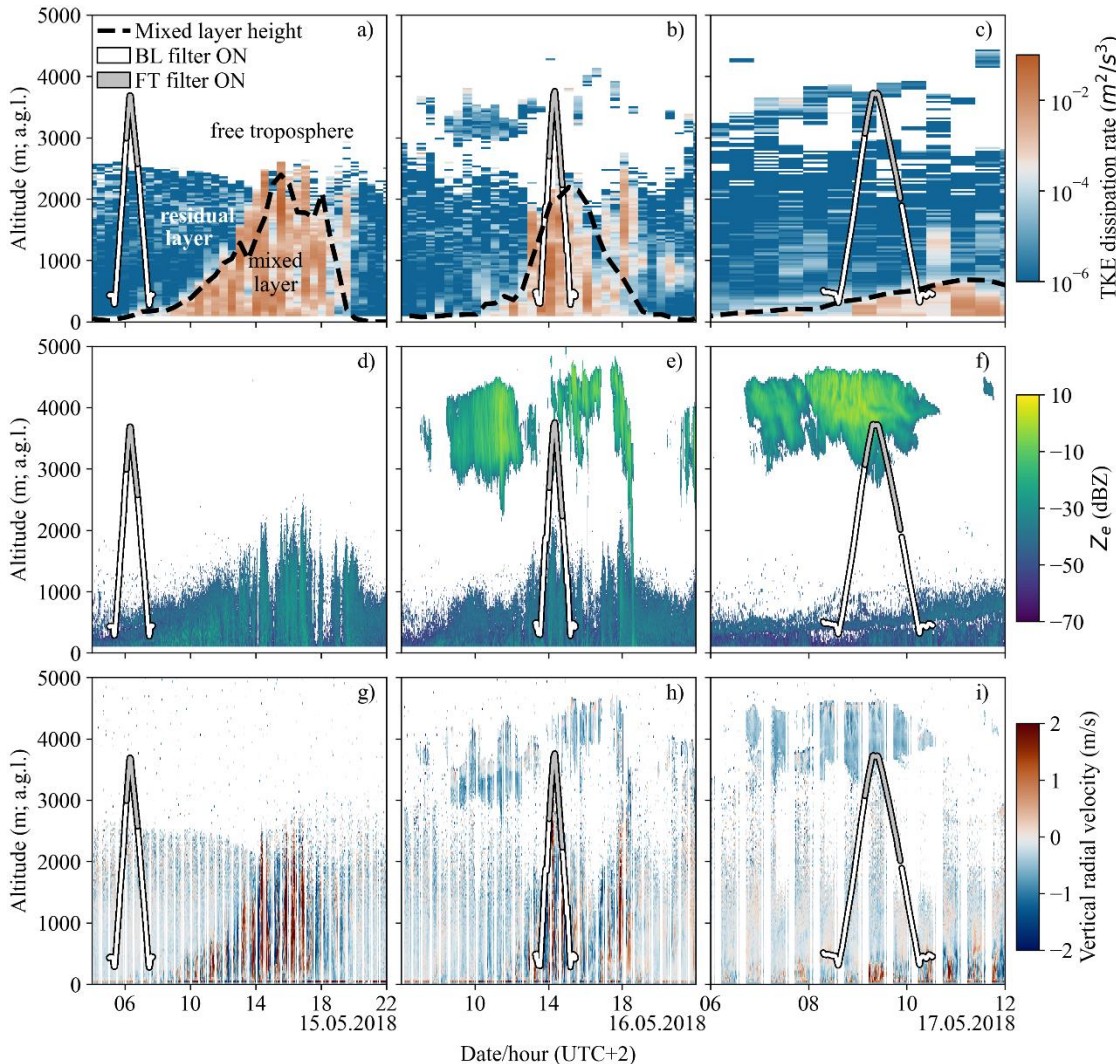

**Figure 10. Evolution of a), d) and g) the boundary-layer height estimated from the TKE dissipation rate measured with the SMEAR II lidar (dashed black line), b), e) and h) the equivalent reflectivity $Z_e$ obtained from the SMEAR II Doppler cloud radar during the three consecutive days of the case study, and c), f) and i) the vertical radial velocity from the SMEAR II lidar. The flight tracks are highlighted together with the status of the filters (BL = boundary layer, FT = free troposphere). All times are given in East European Time (UTC+2).**

Based on the similarities in particle and INP concentrations and size distributions between the ground-level, boundary-layer, and free troposphere measurements on 16 May 2018, we hypothesize that the particles and INPs sampled in the free troposphere are local particles transported from the surface to the free troposphere via local vertical mixing rather than long-range transport. This hypothesis is supported by examining the airmass history of the free troposphere layer sampled during the flight simulated with FLEXPART. In Fig. 11a-b, we present the horizontal distribution of the vertically integrated PES above 3 km and in the lowest 1 km a.g.l., respectively, for air masses arriving at 3 km a.g.l. at SMEAR II on 16 May 2018 at 14:00 (UTC+2). In both cases, the airmasses covered short distances and circulated over Northeastern Europe, similarly to

what was observed in Fig. 6a. Figure 11c displays the vertical distribution of the PES during the 3-day backward simulation for 41 height levels spanning from 50 m to 10 km with a vertical resolution of 250 m. Results show that the air masses spent

very little time below 250 m a.g.l., and therefore are less likely to have accumulated surface particles in transit. However, Fig. 11c shows that the air masses did spend time in the boundary layer, even on the same day as the aircraft measurements. This, together with the similarities in the aerosol population, suggests that the elevated aerosol sampled in the free troposphere originate in the boundary layer. Note that, as a result of the boundary layer influence on the free troposphere, the Schneider et al. (2021) parameterization performs better at reproducing the free tropospheric INP measurements from 16 and 17 May 2018

(Fig. A7) when using the ground-level ambient air temperature.

The process that would cause the ventilation of particles from the boundary layer to the free troposphere remains unclear for now. The presence of clouds during the aircraft measurements on 16 May 2018 (Fig. 10e) could have altered the vertical potential temperature profile (Fig. 9e) and led to radiative cooling at the cloud top, which could in turn drive turbulence. Such turbulence can be seen in the vertical radial velocity data from the SMEAR II lidar on 16 May 2018 as well as on 15 May 2018

(Fig. 10g-h), but do not seem to extend to the free troposphere. There is much less turbulence on 17 May 2018, although the particle concentration and size distribution observed in the free troposphere on that day suggest that the elevated layer is still influenced by the boundary layer.

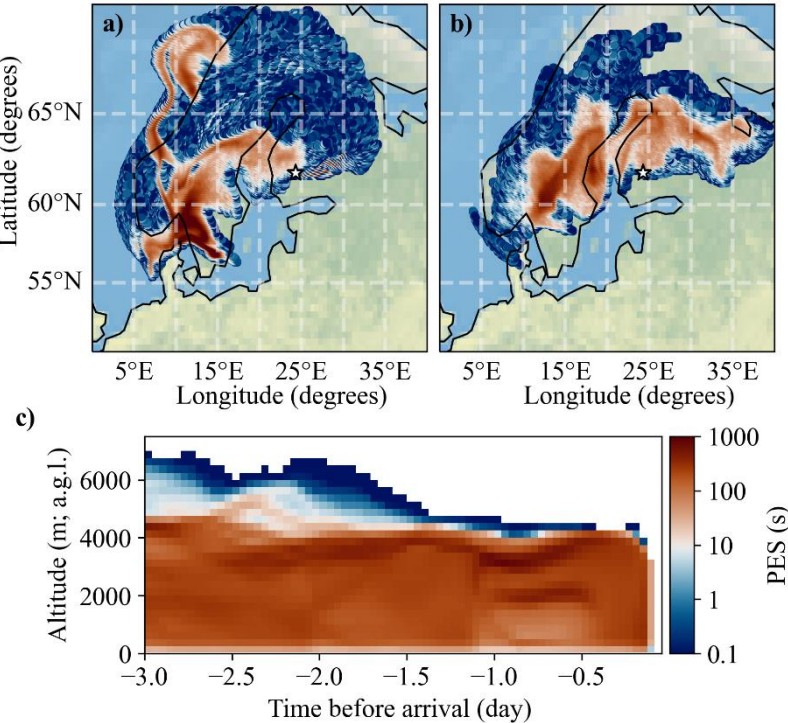

**Figure 11. Airmass origin for the elevated layer at 3 km a.g.l. observed on 16.05.2018 at 14 (UTC+2) in SMEAR II. a) PES summed**
**up for all heights above 3 km a.g.l. for 3 days before arrival at SMEAR II. b) Sum of PES in the lowest 1 km a.g.l. for 3 days before arrival at SMEAR II. c) Vertical distribution of PES for 3 days before arrival at SMEAR II.**

## 4. Conclusions

In this study, we present the first aircraft measurements of INP concentrations above the Finnish boreal forest, and we shed new light on the vertical distribution of INPs above this environment. We found that local surface particles were transported and mixed within the boundary layer aloft through convective mixing, resulting in similar INP concentrations and activated fractions observed at ground level and in the boundary layer. INP concentrations and activated fractions measured in the boundary layer were within the same order of magnitude as those reported by Schneider et al. (2021) for the same period and were best predicted by the parameterization developed in the same study. This further suggested that INPs sampled in the boundary layer primarily originated from the local boreal forest environment rather than long-range transported particles. Although the identity of the INPs sampled in the boundary layer, and whether or not they are dominated by biogenic aerosol similarly to what Schneider et al. (2021) found, has yet to be confirmed, our results suggest that the Finnish boreal forest is the main source of INPs observed in the boundary layer. Future measurements should include additional analysis of the chemical composition and heat sensitivity of the sampled INPs, in a similar manner to what Hartmann et al. (2020), Hill et al. (2016), and Sanchez-Marroquin et al. (2023) have done.

On the other hand, much lower INP concentrations were observed in the free troposphere. The distinct particle number size distributions observed there indicated that different aerosol and INP populations were encountered in the free troposphere and that local surface particles have a weaker influence at these altitudes. The free tropospheric INPs likely resulted from long-range transported particles from different sources, although the analysis of the airmass backward trajectories in the free troposphere did not yield conclusive results due to the limited number of observations. Additional measurements are needed to draw conclusions on the influence of the air mass origin(s) on the INP concentrations and identify the source(s) of INPs observed in the free troposphere.

We showed one case where INP concentrations and activated fractions measured in the free troposphere were higher and within the same order of magnitude as the concentrations observed at ground level and in the boundary layer. We found that, during this flight, the air mass sampled in the free troposphere was influenced by the boundary layer. Although the exact transport mechanism remains unclear, it is possible that particles and INPs were transported to the free troposphere via boundary layer ventilation. Ventilation of the boundary layer into the free troposphere above SMEAR II could likely be caused by convection, turbulent mixing across the capping inversion, or upwards vertical motions of large scale weather systems (e.g., Agustí-Panareda et al., 2005; Donnell et al., 2001). Overall, this finding is of particular importance since INPs in the free troposphere can have longer lifetimes and travel farther, and may therefore expand their range of influence on cloud formation to a regional or global scale.

**Appendix**

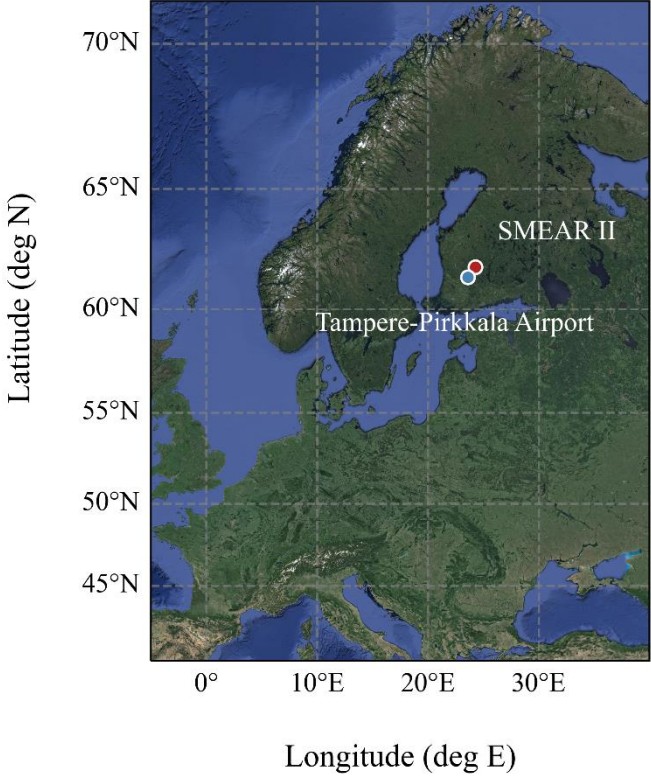

Longitude (deg E)

**Figure A1. Location of the Tampere-Pirkkala airport and SMEAR II with respect to Northern Europe.**

**A1. Background correction of the aircraft filter samples**

The INP concentrations extracted from the aircraft samples were corrected for the INP concentration derived from handling blank filters collected onboard the aircraft. To do so, the INP temperature spectra obtained from the handling blanks were fitted exponentially and averaged to produce a single background curve used for background subtraction (Fig. A2a). Only the

INP concentrations that were at least twice as high as the background curve were considered significant and used in this study. This means that, for a given sample, only the data points meeting this criterion were used, while the data points not meeting the threshold were removed from the analysis (Fig. A2a).

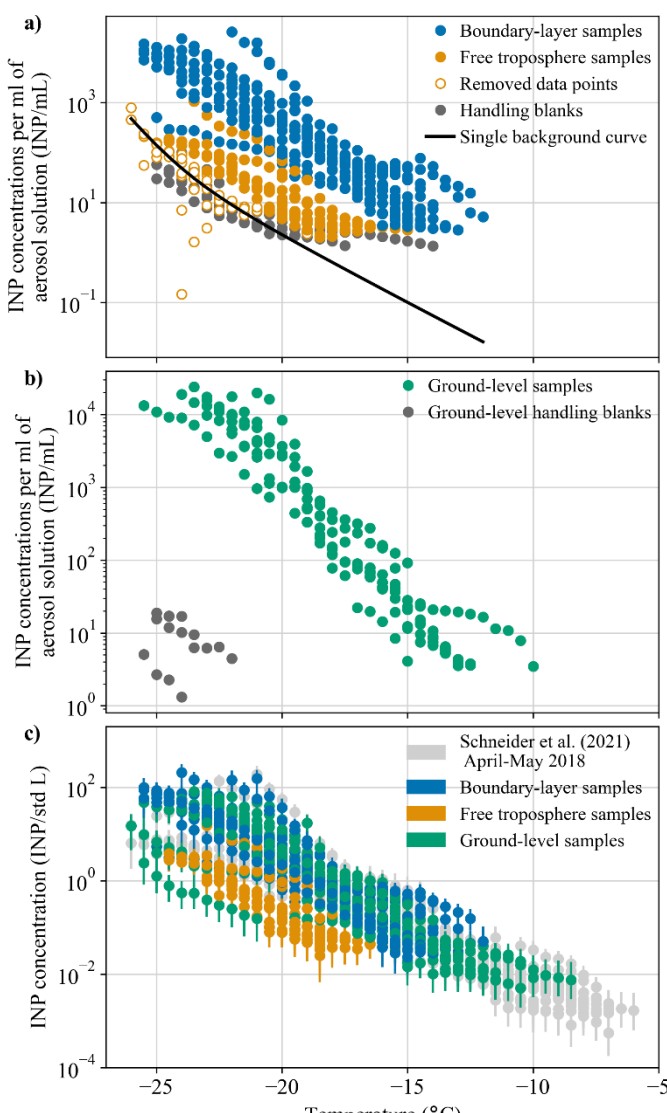

**Figure A2. a)** INP concentrations per ml of aerosol solution from the aircraft samples compared to the background signal derived from the handling blank filters collected onboard the aircraft. **b)** INP concentrations per ml of aerosol solution from the ground-level samples collected at SMEAR II at the same time as the aircraft samples compared to the ground-level handling blank filters. **c)** INP temperature spectra of all the samples collected during the aircraft measurement campaign together with the ground-level data from Schneider et al. (2021) collected in Hyytiälä during April and May 2018. The error bars represent the statistical as well as the systematic error of the INSEKT assay. More details related to the calculation of these error bars is given in Schneider et al. (2021).

**Table A1: Flight campaign overview. The flight times given in the table correspond to the total flight time, including the transit from the Tampere-Pirkkala airport to SMEAR II. All times are given in East European Time (UTC+2). The times of sunrise and sunset were obtained from NOAA (https://gml.noaa.gov/grad/solcalc/).**

| Flight number | Flight date | Flight start time | Flight end time | Apparent Sunrise time | Apparent Sunset time |
|:---:|:---:|:---:|:---:|:---:|:---:|
| 1 | 20 April 2018 | 11:32 | 14:17 | 04:45 | 20:00 |
| 2 | 03 May 2018 | 10:50 | 13:25 | 04:05 | 20:36 |
| 3 | 07 May 2018 | 10:22 | 13:23 | 03:53 | 20:47 |
| 4 | 08 May 2018 | 09:20 | 12:17 | 03:51 | 20:50 |
| 5 | 08 May 2018 | 13:07 | 15:36 | 03:51 | 20:50 |
| 6 | 09 May 2018 | 09:07 | 11:39 | 03:48 | 20:53 |
| 7 | 09 May 2018 | 13:10 | 15:44 | 03:48 | 20:53 |
| 8 | 10 May 2018 | 09:20 | 12:02 | 03:45 | 20:56 |
| 9 | 10 May 2018 | 13:21 | 16:18 | 03:45 | 20:56 |
| 10 | 14 May 2018 | 10:46 | 13:41 | 03:34 | 21:07 |
| 11 | 14 May 2018 | 15:43 | 18:37 | 03:34 | 21:07 |
| 12 | 15 May 2018 | 04:46 | 07:50 | 03:31 | 21:09 |
| 13 | 15 May 2018 | 10:01 | 12:53 | 03:31 | 21:09 |
| 14 | 16 May 2018 | 12:55 | 15:41 | 03:29 | 21:12 |
| 15 | 17 May 2018 | 08:02 | 10:43 | 03:26 | 21:15 |
| 16 | 17 May 2018 | 12:27 | 15:05 | 03:26 | 21:15 |
| 17 | 18 May 2018 | 12:03 | 14:41 | 03:23 | 21:17 |
| 18 | 19 May 2018 | 11:08 | 13:49 | 03:21 | 21:20 |
| 19 | 19 May 2018 | 15:46 | 18:35 | 03:21 | 21:20 |

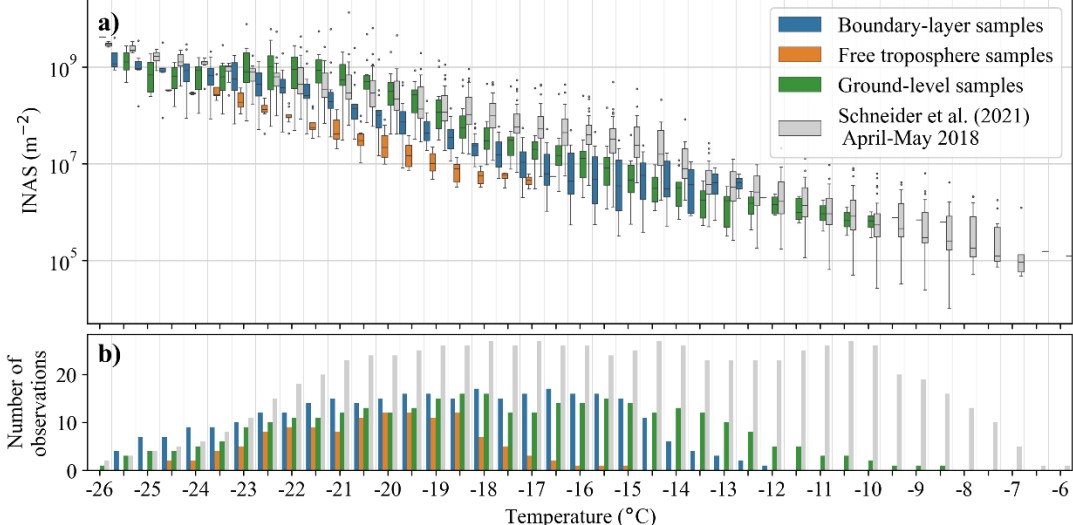

**Figure A3. a) INAS densities as a function of the activation temperature for all the samples collected during the aircraft measurement campaign together with the ground-level data from Schneider et al. (2021) collected in Hyytiälä from 20 April to 19 May 2018. The INAS densities were calculated by normalizing the INP concentration by the aerosol surface area concentration following the method described in Ullrich et al. (2017) and assuming that each INP triggers the formation of one ice crystal. The aerosol surface area**

**concentration was derived from the size distribution measurements of the particles larger than 300 nm obtained from the OPS and the combined DMPS-APS for the aircraft and the ground-level samples, respectively. b) Number of observations for each sample type as a function of temperature.**

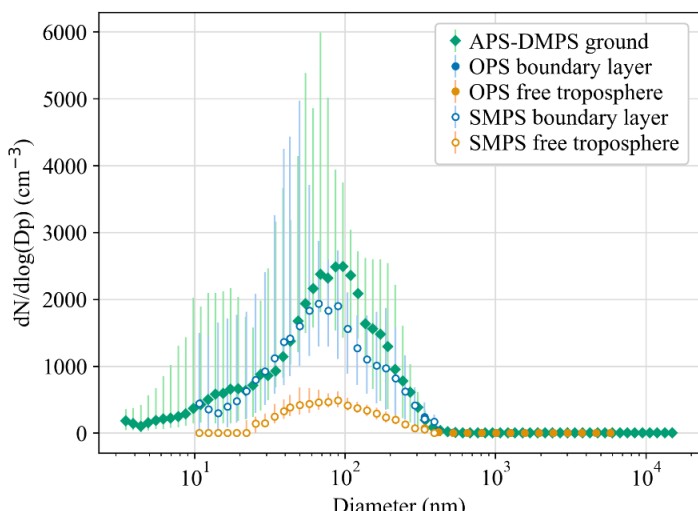

**Figure A4. Median particle number size distributions calculated from ground-level measurements (SMEAR II APS and DMPS) as well as boundary layer and free troposphere measurements (aircraft SMPS and OPS) over the 19 flights of the campaign. The error bars represent the 25th and 75th percentiles.**

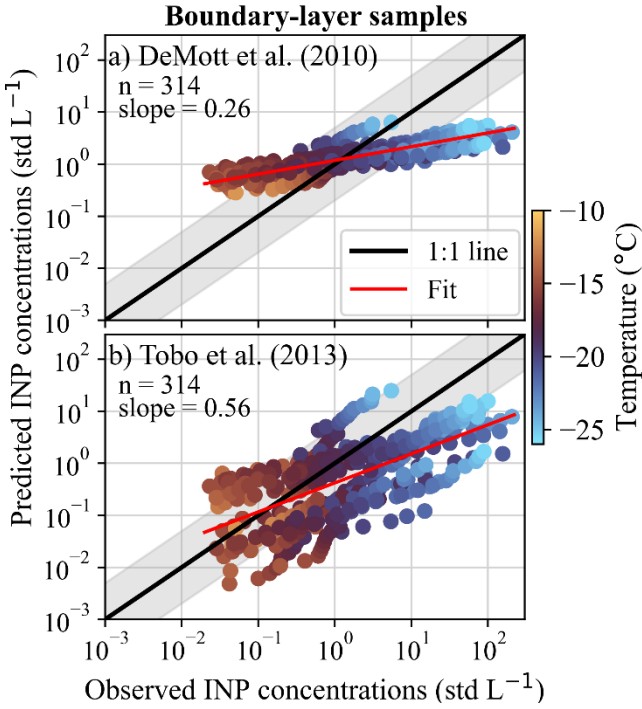

**Figure A5.** Comparison between the INP concentrations observed in the boundary layer and the INP concentrations predicted using the parameterizations from a) DeMott et al. (2010), and b) Tobo et al. (2013) using the aircraft OPS data. DeMott et al. (2010) reproduces 60 % and 32 % of the data points within a factor of 5 and 2, respectively. Tobo et al. (2013) reproduces 49 % and 17 % of the data points within a factor of 5 and 2, respectively. The black solid line represents the 1:1 line while the grey shaded area indicates a range of a factor of 5 from the 1:1 line. The red solid lines show a linear regression fit through the logarithmically transformed data points. The slope of the fit and the number of data points used is shown in each panel.

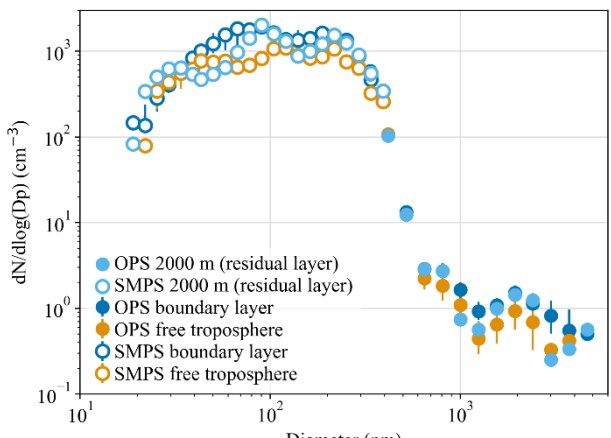

**Figure A6. PNSD measured on 17 May 2018 at the highest point reached in the free troposphere and in the first meters of the residual layer.**

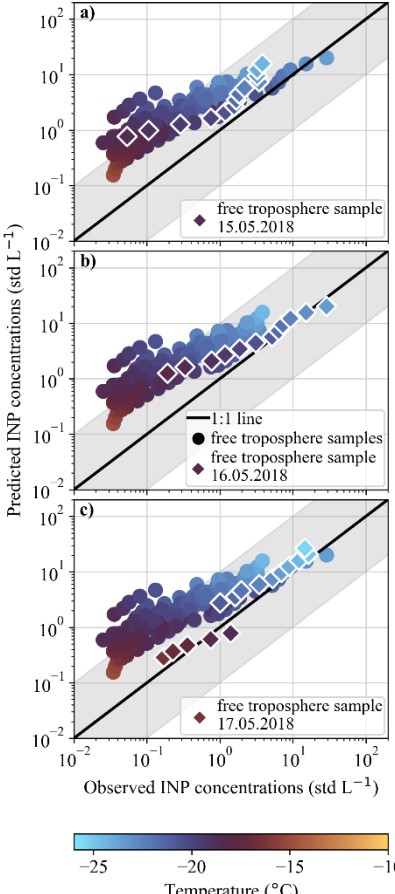

**Figure A7. Comparison between the observed and the predicted INP concentrations calculated using the parameterization from Schneider et al. (2021) for the free troposphere samples collected on the three consecutive days of the case study. In each panel, the diamond markers represent a specific free troposphere sample sampled on a) 15 May, b) 16 May, and c) 17 May 2018, plotted on top of all the free troposphere samples collected during the campaign. The black solid line represents the 1:1 line while the grey shaded area indicates a deviation of 1 order of magnitude from the 1:1 line The parameterization from Schneider et al. (2021) was calculated using the ground-level ambient air temperature measured at 4.2 m a.g.l. and averaged over the sampling time of each sample.**

**Data availability**

The INP and aircraft data presented in this article will be available upon publication with the following DOI: 10.5281/zenodo.10975295. The aerosol and meteorological data from SMEAR II can be accessed at https://smear.avaa.csc.fi/ (Junninen et al., 2009). The ground-based 24-hour measurements from Schneider et al. (2021) are available via the KITopen data repository under https://doi.org/10.5445/IR/1000120666 (Schneider et al., 2020). The data from Barry et al. (2021) provided by NCAR/EOL under the sponsorship of the National Science Foundation are available at https://doi.org/10.26023/A7KM-DDNK-MX0T (DeMott et al., 2020). The data from Conen et al. (2022) are given in the Appendix of their publication. The data from Twohy et al. (2016) are available in their Supplements. The data associated with the publication from Sanchez-Marroquin et al. (2021) is available from the University of Leeds at https://doi.org/10.5518/979. The INP data from Hartmann et al. (2020) are available at https://doi.pangaea.de/10.1594/PANGAEA.899635. The data from Petters and Wright (2015) are available in their supporting information. The data from Schrod et al. (2017) can be accessed through the BACCHUS database under: http://www.bacchus-env.eu/in/info.php?id=72.

**Author contribution**

Markku Kulmala and Tuukka Petäjä developed and scientifically lead the research program at SMEAR II station. Jonathan Duplissy conceived the idea of the study. Zoé Brasseur, Janne Lampilahti, Markus Lampimäki and Pyry Poutanen prepared and set up the instruments onboard the aircraft and conducted part of the aircraft measurements. Julia Schneider analyzed the INP filter samples. Janne Lampilahti and Pyry Poutanen processed the aircraft data. Carlton Xavier carried out the FLEXPART simulations. Ville Vakkari analyzed the SMEAR II HALO Doppler lidar data. Dmitri Moisseev analyzed the SMEAR II Doppler cloud radar data. Zoé Brasseur and Julia Schneider conducted the data analysis. Zoé Brasseur wrote the manuscript with contributions from Jonathan Duplissy, Katrianne Lehtipalo, Ottmar Möhler, Kristina Höhler, Erik S. Thomson, Markus Hartmann, Christina J. Williamson, Victoria A. Sinclair, Ville Vakkari, Dmitri Moisseev, Carlton Xavier, Pyry Poutanen, Markus Lampimäki, and Markku Kulmala.

**Competing interests**

At least one of the (co-)authors is a member of the editorial board of Atmospheric Chemistry and Physics

**Acknowledgement**

The authors would like to thank Erkki Järvinen and the pilots at Airspark Oy for operating the research airplane, and we are grateful for their hospitality and helpfulness. We thank the technical staff of the Hyytiälä Forestry Field Station for their help throughout the HyICE-2018 campaign. Antti Mannisenaho, Yusheng Wu, Paula Hietala, Anna Franck, and Maija Peltola are

acknowledged for their help with the aircraft measurements. The Scientific color maps described in Crameri et al. (2020)are
used in this study to prevent visual distortion of the data and exclusion of readers with color vision deficiencies. The color
maps are available at: Crameri, F. (2018), Scientific colour maps, Zenodo, doi:10.5281/ zenodo.1243862.

**Financial support**

We acknowledge the following projects: ACCC Flagship funded by the Academy of Finland grant number 337549, Academy
professorship funded by the Academy of Finland  (grant no. 302958), Academy of Finland projects no. 325656, 311932,
334792, 316114, 325647, 325681, 333397, 328616, 357902, 345510, 347782, "Quantifying carbon sink, CarbonSink+ and
their interaction with air quality" INAR project funded by Jane and Aatos Erkko Foundation, "Gigacity" project funded by
Wihuri foundation,  European Research Council (ERC) project ATM-GTP Contract No. 742206, and European Union via
Non-CO2 Forcers and their Climate, Weather, Air Quality and Health Impacts (FOCI), and CRiceS (No 101003826), RI-
URBANS (101036245), EMME-CARE (856612) and FORCeS (821205). The Technology Industries of Finland Centennial
foundation via urbaani ilmanlaatu 2.0 -project is gratefully acknowledged. University of Helsinki support via ACTRIS-HY is
acknowledged. Dmitri Moisseev was supported by the funding from Horizon Europe programme under Grant Agreement No
101137680 via project CERTAINTY (Cloud-aERosol inTeractions & their impActs IN The earth sYstem). This research has
been supported by the Horizon 2020 (grant nos. ACTRIS-2 654109, ACTRIS PPP 739530, ACTRIS IMP 871115) and the
Helmholtz Association (grant no. 120101). Erik S. Thomson was supported by the Swedish Research Council VR (2020-
03497).

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
