# Peer review of "Vertical distribution of ice nucleating particles over the boreal forest of Hyytiälä, Finland"

_EGUsphere, 2024_

## Editor Comment (EC1)

Vertical distribution of ice nucleating particles over the boreal forest of Hyytiälä, Finland

Zoé Brasseur et al.,

This manuscript describes ice nucleation measurements collected in three different atmospheric compartments; ground level, boundary layer, and free troposphere. Boundary layer and free troposphere measurements were made aboard a CESSNA aircraft. The duration of the sampling period ranged from the 20 April until 20 May 2018, and involved 19 separate flights. Ground based measurements took place at the SMEAR II station in Hyytiala. The aircraft was equipped with filter measurements (for INP analysis) and with SMPS and OPS measurements for aerosol size and number concentration

Several meteorological and remote sensing measurements were used to determine the development of the boundary layer each day. This information was later used to interpret the variability INP concentrations as a function of altitude.

Measurements are compared to previous studies at the same site over a longer period of time. These measurements and the subsequent aerosol measurements are used to develop an updated parameterization that is compared to existing parameterizations from a number of different studies.

There are few studies capable of providing measurements on the vertical profile of INP measurements and therefore this study presents a very unique and valuable data set. The manuscript is well written and the graphic are of very good quality.

**Please find below some general comments and suggestions:**

**Data availability:**

It would be appreciated to provide accessible data to the reviewers (as is stipulated in the policies of ACP).

**Introduction:**

The presentation and discussion of the parameterizations is an important part of this manuscript. A short discussion on the need for parameterizations in the introduction would be useful, and to present why having multiple variable parameterization's is useful compared to single parameterization's (currently used in cloud models). Can cloud models integrate these more complex parameterizations?

**Methodology:**

- Were any samples analyzed immediately after collection and again post freezing. What kind of impact is the freezing cycle expected to have on the ice nucleation activity of aerosol particles?
- Line 189: Can the authors rephrase this sentence. What exactly was passed through the filter and what was the purpose of using this filter. "*First, the sampled aerosol particles were washed from the filter membrane into a solution using Milli-Q purified water (18.2 MΩ.cm), which was passed through a 0.1 µm Whatman syringe filter.*"
- What is the volume of liquid used in the sample wells of INSEKT, and how many wells are used? Was it possible to perform multiple runs on the same samples to determine the reproducibility of the filters? What impact of biological aerosol is thought to have on these samples? Schneider et al., performed a heat treatment of the samples, and observed a significant fraction of biological INPs. Do the authors suspect that this might also be the case in these samples, and how would this vary with altitude?

- Can the authors provide Fig. A1 in the same units (INP stdL) as the data presented elsewhere. It would be easier for the reader if Figure A1 and A2 are combined in the same figure a) and b).
- For the air mass backtrajectories, was the impact of airmass history (precipitation, and or cloud activation, from ECMWF calculations) considered in this analysis.

**Results:**

- Can aerosol mixing state be inferred from the size distribution measurements? What impact does aerosol mixing state have on INP properties?
- What causes the difference in the onset of freezing between Schneider et al., 2021 and the ground level samples (shown in figure 4). What are the differences between the Schneider sampling set up at SMEAR II and those in this work? The authors say that they were sampling behind a TSP inlet, however Schneider was sampling behind à PM10. What is the inlet height at the ground station?
- Figure 4b, would it not be more correct to label this normalized INP / particle number concentration. The axes being labeled Activated fraction, is misleading.
- Can the authors provide more details about the data points in Figure 4 (Median, percentiles etc..).
- Since measurements of the full size distribution were available why was the number of INP per aerosol surface area not calculated? This is a pertinent measurement of INP activity and would be possible to put it in context with a number of other studies.
- In the individual plots (shown in Figure 9), we observe very different distributions of INP concentrations as a function of temperature. For example at ground level on the 15[th] there was a first freezing mode until -18C then a sudden increase in freezing until -20C and then a gradual increase until the end. Whereas in the boundary layer the INP concentrations appear more homogenous with a consistent rise in INP as a function of temperature. Likewise in the FT measurements there appears to be a similar distribution as the BL. Are these changes in INP spectra a result of dilution steps (as discussed for 9c) or is there additional information that can be extracted? Are there different 'slopes', or freezing modes in the data.
- For the jumps in data points observed for the FT samples collected on the 17[th], that the authors state is a result of a dilution step, should the reader focus on the trend in the FT sample after the -20°C of before the -20°C. Are measurements valid in both cases?

- The average INP plots over the full period were illustrated. As expected a high spread in the data points is noted. The authors showed that there was significant changes in weather (Figure 3) over all the flights.

  o Can the variation in the INP concentrations be explained by other variables than aerosol size, such as the temperature and pressure, and also aerosol chemistry (at least from ground based measurements).

  o Can aerosol chemistry (other than biogenic sources) be mentioned, at least for the ground based measurements (from measurements available at the SMEAR station), or those measurements already available in the Schneider paper. Was there any variability observed during the sampling period discussed in this manuscript.

- **Figure 9:** The authors mention that there were clouds present between 3000 and 4000 m on the flight of the 17[th]. Was the aircraft sampling inlet adapted to sample could droplets?
- **Figure 7:** For the Schneider parameterization (in both the BL and the FT), it seems that there are only a small number of points that are pulling the fit away from the 1:1 line.
    - Do these points correspond to a single flight? If so, is there something particular about this flight?
    - Why do the authors not provide a test for the ground based measurements?

- In figure 7:  Is it possible to apply this newly developed parameterization's to the ground based, boundary layer data, and also to Schneiders data. It would be interesting to determine if this parametrization can be used in other environments or if it is only suitable for FT measurements.

- In Table 1, it would be useful to also include the fitting values of the updated parameterization included here. Those that are listed at the end of the paragraph.
- In the development of the parametrization, what methods were used to find the optimum values for the coefficients? Were these calculated only using the FT samples?

---

## Author Comment (AC1)

The manuscript by Brasseur et al focusses on the vertical distribution of INP over boreal forest in Finland and has been carried out by aircraft measurements during spring 2018 above Hyytiälä. The manuscript is well-written and includes new and important data. The topic is important for atmospheric scientists and might be published in ACP after some minor corrections.

**Response**: We thank the referee for their useful comments which helped improve the manuscript. We include our responses below in the context of individual comments.

**Comments**:

Introduction: The topic has been well introduced and the relevant new literature has been cited. However, the authors might mention their own paper Vogel et al. which is under discussion as well. Also, the older literature at similar locations deserves attention (e.g. Prenni 2013). The authors mention the ice nucleation temperature of desert dust being below -15°C but I miss a statement regarding the biological INPs and INMs which are below -2°C depending on the origin. In this regards the authors might also mention that dust could be a carrier, coated with bio INMs, which turns them into bio INPs.

**Response**: We propose adding a mention to the Vogel et al. (2024) paper line 112:

"Finally, Vogel et al. (2024) presented ground-based measurements conducted with the Portable Ice Nucleation Experiment (PINE) below -24 °C and found moderate correlations between INP concentrations and concentrations of particles larger than 0.5 µm as well as concentrations of fluorescent aerosol particles, hinting at a possible biological source of INPs active below -24 °C."

In addition, we propose mentioning previous studies conducted at similar locations (forested environments) line 93:

"These results agree well with previous studies conducted in similar forested environments. Prenni et al. (2009) for example showed that INP concentrations and abundance in a pristine rainforest of the Amazon basin could be partly explained by local emissions of biological particles. Huffman et al. (2013), who performed measurement in a semi-arid pine forest of North America, found a strong correlation between fluorescent biological particles and INPs during rain events. Similarly, results presented in Prenni et al. (2013) suggest that biological particles represent a significant portion of rain-generated INPs measured at a forested site in Colorado. Tobo et al. (2013) conducted measurements in a midlatitude ponderosa pine forest ecosystem in Colorado and found significant correlations between INP concentrations and the concentration of ambient fluorescent biological aerosol particles. Finally, Iwata et al. (2016) carried out measurements near forested mountain slopes in Japan and found that biological particles played an important role as INPs for temperatures warmer than -22 °C, especially during rainfall events. However, all these observations were carried out at ground level and did not examine the transport of such INPs to higher altitudes."

Concerning biological INPs, we propose modifying lines 47-50 to:

"Biological aerosols are considered another widely present type of INPs (O′Sullivan et al., 2015; O'Sullivan et al., 2018; Wex et al., 2019;). Although their global emissions are lower than dust, they can form ice at relatively warmer temperatures depending on the nature of the bioaerosols (Després et al., 2012). For example, the bacteria *Pseudomonas Syringae* is a very efficient INP at temperatures as warm as -2 °C (Maki et al., 1974; Joly et al., 2013). In addition, biological particles, including bacteria, have been found in dust aerosols, possibly enhancing their ice nucleation activity (Conen et al., 2011; Meola et al., 2015; Barr et al., 2023)."

Methods: The description is very detailed. However, I have remarks regarding the figures:

Figure 1a: The map is very pale and the scripts are small. I would recommend to add a general map where the location on the European continent is documented.

**Response**: Following the referees' comments, Fig. 1a was updated to a 3D plot to highlight the flight tracks more clearly. A figure with the location of the measurement site with respect to the European continent was added in the Appendix. The updated figures are attached at the end of this document.

Figure 2: The color code is missing. The x-axis might be labeled "day time" or "position of the sun"

**Response**: We added a legend with the color code and a label for the x-axis. The updated figure is attached at the end of this document.

Results: Figure 3: Please explain the calculation of the TKE dissipation rate from the Doppler lidar in more detail.

**Response**: Thank you for your comment. We propose modifying the text lines 245-251 to the following:

"After the flights, data from a Halo Photonics Stream Line scanning Doppler lidar located at SMEAR II was used to estimate the limit between the boundary layer and the free troposphere, for comparison with the aircraft measurements. The Halo Doppler lidar was configured with vertically-pointing stare and conical scans (i.e., with velocity azimuth display (VAD) scans at 30 ° elevation angle) repeating every 30 minutes. Additional scans during the 30-minute scan cycle were not used in this study. The range resolution of the lidar is 30 m, with a minimum range of 90 m a.g.l.. More details on the Doppler lidar at SMEAR II can be found in e.g., Hellén et al. (2018). The data was post-processed following Vakkari et al. (2019): horizontal winds were retrieved from the VAD scans and the variance of vertical wind velocity was calculated from 12 consecutive vertical stare measurements. The instrumental noise contribution to the observed variance of vertical wind velocity was estimated from the post-processed signal-to-noise-ratio according to Pearson et al. (2009) and subtracted before calculating turbulent kinetic energy (TKE) dissipation rate profiles according to the method by O'Connor et al. (2010)."

Figure 5a: add ticks on the y-axis for every order of magnitude.

**Response**: We added ticks for every order of magnitude on the y-axis. We attach the updated figure at the end of this document.

Figure 6a: add a size bar, explain the color code in more detail

**Response**: We added a title to the legend and modified the figure caption to improve clarity regarding the color code. We attach the updated figure at the end of this document. However, we would rather not add a scale bar in Fig. 6a as it may be inaccurate due to the distortions that can occur with map projections at this scale. We believe that the gridlines representing the latitudes and longitudes should be enough to provide context to the map used.

Figure 7: add ticks on the x- and y-axes for every order of magnitude

**Response**: We added ticks for every order of magnitude on the x and y axes. We attach the updated figure at the end of this document.

Figure 11a and b: add size bar. Better explain PES.

**Response**: As in Fig. 6a, we would prefer not to add scale bars to Fig. 11a and b since adding a straight line to represent the scale would be inaccurate due to distortions in the projection. Concerning the PES, we propose adding the following text to the methods line 264:

"FLEXPART was used to calculate the potential emission sensitivity (PES) fields, where PES is the response function of a source-receptor relationship which estimates the potential source contributions for a given receptor (in this case the measurement site SMEAR II). The PES is therefore proportional to the residence time of the air mass in a specific grid cell, and it was calculated in units of seconds. High values of PES indicate source regions where emissions are likely to significantly impact the tracer concentrations at the receptor (Seibert and Frank, 2004; Stohl et al., 2005; Pisso et al., 2019)."

Conclusion: The conclusion reads more like a summary. I miss a real discussion and conclusion from the results along these questions: What do we learn regarding the emission of INP and their transport from the forest ecosystem, above canopy, into the free troposphere? Can you speculate about the transport mechanisms? How do wind and humidity influence these potential processes?

**Response**: Following the comments from Referees 1 and 2, we have shortened and focused the conclusions. Here is the updated text:

[revised manuscript text omitted]

---

## Author Comment (AC2)

The "Vertical distribution of ice nucleating particles over the boreal forest of Hyytiälä, Finland" by Brasseur et al. is an interesting study investigating how INPs vary with height in a region that could release a lot of INPs and is understudied. Aircraft measurements of INPs are particularly useful, because the majority of studies are ground-based and thus cannot usually provide measurements at the heights where clouds can form. The authors present both INP and aerosol data from several flights, which they use to normalize the spectra for insight into vertical efficiency. I especially thought the case study section was strong and gave convincing evidence for the possibility of periodic transport from the surface to the free troposphere but that most of the time it is different. It would be most interesting to investigate further (in the future) the potential mechanisms for the transport.

Despite the positives of this work, there are several things I believe can be improved upon, both in the figures and text. The main things I note are being clearer about the blank corrections, as they matter significantly for the free tropospheric samples; being extremely careful about not overstating claims: I don't think some of the conclusions are supported by the data presented (as worded). This happens throughout the manuscript. I also thought the number of studies for comparing to previous work could be improved. In general, I think the manuscript reads too long, and could be tightened to improve its readability. I fully intend and hope my comments are helpful for the coauthors and offer a useful perspective. I would recommend it for publication, but only after these comments are given consideration and the claims in the text are represented better.

**Response**: We thank the referee for their feedback on the manuscript, which has greatly improved due to the referee comments and suggestions. Our responses are given below (in blue font) in the context of individual comments.

**Specific comments:**

Introduction: Should add Levin et al. (2019) as they made vertical measurements of INPs from the surface to free troposphere over California, for comparison. They found increased concentration with height.

Levin, E. J. T., DeMott, P. J., Suski, K. J., Boose, Y., Hill, T. C. J., McCluskey, C. S., Schill, G. P., Rocci, K., AlMashat, H., Kristensen, L. J., Cornwell, G. C., Prather, K. A., Tomlinson, J. M., Mei, F., Hubbe, J., Pekour, M. S., Sullivan, R. J., Leung, L. R., and Kreidenweis, S. M.: Characteristics of ice nucleating particles in and around California winter storms, J. Geophys. Res.-Atmos., 124, 11530–11551, https://doi.org/10.1029/2019JD030831, 2019.

**Response**: Thank you for pointing out missing literature. We propose adding the following text line 74:

"Similarly, Levin et al. (2019) observed an increase in INP concentration and in the fraction of total aerosol particles capable of ice nucleation from the surface up to approximately 7 km above sea level in wintertime in California, and suggested that pollution aerosols near the surface were poor sources of INPs."

Lines 103-112: Much of this should go in the methods and is distracting from the main message of the introduction.

**Response**: We agree with the referee that some of the details might be distracting from the main message of the introduction. However, we also believe that the summary of the previous findings from the HyICE-2018 campaign and their link to this manuscript belongs more in the introduction than in the methods. Therefore, we suggest rephrasing the paragraph lines 103-112 to:

"In this study, we present filter-based measurements of INPs conducted at ground level and aloft in the boundary layer and free troposphere (up to an altitude of 3.5 km) in and above a Finnish boreal forest during spring 2018. The measurements were organized in the framework of a larger ice nucleation measurement campaign, called HyICE-2018, which took place at the Station for Measuring Ecosystem–Atmosphere Relations (SMEAR II; Hari and Kulmala, 2005) in Hyytiälä, Finland and presented in details in Brasseur et al. (2022). Results from the HyICE-2018 are also available from Paramonov et al. (2020), who presented ground-based INP concentrations measured with the Portable Ice Nucleation Chamber (PINC) during the first part of the campaign. In addition, the study from Schneider et al. (2021) extended their measurements for more than one year after the HyICE-2018 campaign and focused on immersion freezing INPs measured with the Ice Nucleation Spectrometer of the Karlsruhe Institute of Technology (INSEKT). They showed that the surface INP concentrations have a clear seasonal cycle that appears linked to the abundance of boreal biogenic aerosol. Building from these previously published results, the objective of this study is to describe the vertical variability in INP concentrations from ground level to the free troposphere above the Finnish boreal forest environment. To do so, we use instrumentation installed both onboard a measurement aircraft and at the SMEAR II measurement site, which allows for comparison between INP measurements and simultaneous measurements of many particle and meteorological variables."

Figure 1a: Map is hard to read, but I like the colored flight track. Can you improve the resolution (and maybe spatial extent)?

**Response**: Based on the Referees' comments, Fig. 1a was updated to a 3D plot to highlight the flight tracks more clearly. A map depicting the measurement site location in northern Europe is added to the Appendix. The updated figures are attached at the end of this document.

Line 160: A place with the basics of filter collection is needed: how many of each locational type were collected?

**Response**: We propose adding the following information line 199:

"In total, from the 19 measurement flights, 18, 16, and 16 filters were collected in the boundary layer, in the free troposphere, and at ground level, respectively."

Line 162: Please specify the length of soaking in 10% $H_2O_2$ and number of water rinses

**Response**: We added the missing information line 162:

"Before sampling, the filters were pre-cleaned by soaking them with 10 % $H_2O_2$ for 10 minutes. Afterwards, they were rinsed three times with deionized water that was passed through a 0.1 μm Whatman syringe filter."

Line 193/Figure A1: Were the blank samples corrected in INPs/mL suspension space? I don't think that is the unit you want to correct in if your ground samples were resuspended in 8 mL and boundary layer/free troposphere samples were resuspended in 5 mL, as that number is a function of resuspension volume. I think it is necessary to correct in total INPs/filter (multiply by resuspension volume), unless some blanks were resuspended in 8 mL to correct for the ground samples, and other blanks were resuspended in 5 mL. However, if everything was resuspended in 5 mL for your data, it would be ok and please make that clearer.

It would be helpful to indicate on Figure A1 which samples you ignored from being within your threshold, and you could also indicate in the text the percentage of filters you were able to keep. In the text, I would suggest providing more detail about the background corrections, and not only refer to previous literature (even though I know it is a related study). The reason is that often free troposphere/boundary layer airborne filters as you know are very close to the limit of detection, and so blanks can play a very big role in the answers and thus the conclusions that are drawn. Additional questions that I have, for example, are did you average the blanks and create a regression to subtract? For the samples that you adjusted because they were within a factor of two of the blanks, were you able to keep some points or did you remove the entire spectra? It would be great to indicate the points on a plot (Figure A1 for example) that were measured on the INSEKT but did not pass your criteria. I think it would be worthwhile to spend time to make this clearer. I think the criteria is acceptable, but at this point it is not repeatable.

**Response**: We thank the referee for their useful comment on the background correction. Both the samples and the handling blanks from the aircraft were suspended in 5 ml, thus the correction was done in the unit of INPs per ml. We modified the information line 189 to improve clarity:

"For this reason, and to enhance the INP content in the sample solution, the volume of filtered nanopure water was reduced from 8 to 5 ml for the samples collected onboard the aircraft."

And lines 193-197:

"The INP concentrations reported here have been corrected for the background freezing levels of filtered nanopure water. The INP concentrations extracted from the aircraft samples were further corrected for the INP concentration derived from handling blank filters, which were collected onboard the aircraft without ambient air flowing through the membranes. Then, as the INP concentrations measured from the aircraft filters were rather low and close to the background signal derived from the handling blank filters (Fig. A2a), only the INP concentrations that were at least twice as high as the average background INP concentrations were considered significant and used in this study. More information concerning the handling blank correction can be found in Appendix A1. The INP concentrations extracted from the ground-level samples were well above the INP concentration derived from ground-level handling blank filters (Fig. A2b) and were therefore not corrected further."

We also add a short section in the Appendix to describe the procedure followed for the background correction:

"The INP concentrations extracted from the aircraft samples were corrected for the INP concentration derived from handling blank filters collected onboard the aircraft. To do so, the INP temperature spectra obtained from the handling blanks were fitted exponentially and averaged to produce a single background curve used for background subtraction (black line in Fig. A2a). Only the INP concentrations that were at least twice as high as the background curve were considered significant and used in this study. This means that, for a given sample, only the data points meeting this criterion were used, while the data points not meeting the threshold were removed from the analysis (Fig. A2a)."

Finally, we updated Fig. A2a to include the single background and the data points that were removed. We added Fig. A2b which shows the ground-level data compared to the ground-level handling blanks, and we moved the previous Fig. A2 to Fig. A2c to regroup all the subplots together. The updated figure A2 is attached at the end of this document.

Figure 2: This is a very nice figure and is helpful to a general audience.

**Response**: We thank the referee for their positive feedback.

Paragraph beginning with Line 302: The ice onset is not helpful here because as you state, the volumes are very different. I would remove it, or if you keep it in, also qualify the free troposphere portion at the end as the volumes were the shortest (I understood 7 LPM for 1 hour average).

**Response**: We would like to keep the mention of the ice onset and thus added more information concerning the free troposphere samples as suggested by the referee:

Line 318: "[…] They also have an ice onset temperature colder than any other measurements shown in this study (approximately 4.5°C, 7°C and 10°C colder than the ice onset temperatures of the boundary-layer, ground-level and 24-hour measurements from Schneider et al. (2021), respectively). As mentioned previously, this is likely due to shorter sampling times used for the free troposphere samples ($\approx$ 60 minutes)."

Figure 4: I would suggest you have a criteria for plotting/not plotting the histograms (at the cold end) based on the number of observations (e.g. >50%). I understand and appreciate you including the number of observations, but as most apparent in the ground histograms, the colder observations will be biased low based upon the more concentrated samples needing more dilution.

**Response**: We would prefer showing the entire dataset and not removing any data. We agree with the referee that some of the colder observations are biased low due to the limited number of data points at these temperatures, which is why we included the number of observations in Fig. 4c. To avoid confusion related to this matter, we propose mentioning this in the main text, line 305:

"Note that representing the data in this way might introduce some bias when the number of observations used to calculate the boxplots is more limited, for example at colder temperatures

where some of the ground-level INP concentrations appear to be decreasing with decreasing temperature. The number of observations for each sample type as a function of temperature is highlighted in Fig. 4c."

Lines 334-337: The fact that the activated fraction brings the free troposphere closer to the rest of the observations than INP concentrations alone, would there be an argument for that making them more efficient (relatively speaking: still "less efficient" overall) as now some of the histograms overlap (especially with the previous study) over the range with many observations? It is unexpected to me that they are closer, and is an important finding to highlight more, even if it is not within the main message of the paper.

**Response**: We agree with the referee that this is an important observation that should be highlighted. We propose rephrasing lines 334-337 to:

"Figure 4b shows that there is more overlap between the activated fraction from all sample types compared to the INP concentrations shown in Fig. 4a. The activated fraction from the ground-level and boundary-layer samples are within the same order of magnitude, while the activated fraction from the free troposphere samples is overall lower, with some overlap with the ground-level samples below -20 °C. This suggests that, even though particles sampled in the free troposphere are overall less efficient INPs, there are a few cases where the free tropospheric INPs are as efficient as those sampled at ground level. These specific cases are further discussed in section 3.7."

Line 376-379: How do you reconcile the similarity of the activated fractions of the free troposphere to the Schneider et al. study (especially below -20) with this statement? INP concentration speaking, I agree with your statement. Lines 382-383 are affected as well.

**Response**: The activated fraction from the free troposphere samples is brought closer to the rest of the observations due to the few cases where the INP concentration (and activated fraction) observed in the free troposphere was higher. These cases are discussed in section 3.7, where we show that the free troposphere can sometimes be influenced by the boundary layer (itself influenced by the surface). We propose rephrasing lines 382-383 to:

"On the other hand, the lower INP concentrations measured in the free troposphere are most likely due to the lower particle concentrations encountered there combined with the fact that the free tropospheric particles might be less efficient INPs, as suggested by the overall lower activated fraction (Fig. 4b). In addition, the differences observed in the size distribution pattern suggest that the aerosol populations present in the free troposphere are different than those encountered in the boundary layer and at ground level. It is likely that these particles, and thus the INPs, were transported from distant regions via long-range transport, as discussed in section 3.4. Nevertheless, as mentioned previously, the activated fraction from the free troposphere samples sometimes overlap with the rest of the observations, in particular with the ground-level measurements from Schneider et al. (2021) and at temperatures below -20 °C. This shows that there are some cases where the activated fraction of the free troposphere samples is higher and similar to those observed at ground level. Such observation raises the question of whether surface particles might influence

the free troposphere locally in some way. This question is further investigated in section 3.7, where we focus on the flights with the highest INP concentrations observed in the free troposphere."

Paragraph starting with Line 417: This paragraph is wordy and doesn't really present anything new. A point of needing more samples would be sufficient in the conclusions. I would suggest to trim/remove this. I do like the HYSPLIT analysis in this general section.

**Response**: We agree with the referee and propose removing the paragraph and moving part of it to the conclusions.

Line 444: Define CFDC at first use

**Response**: We added the acronym definition at its first use line 444.

Lines 447-450: There is not a sufficient explanation on why the aircraft OPS data was not used for all aircraft samples, even if you are assuming the air is similar enough. I would at least include that representation in the supplemental information for transparency, maybe as an additional figure or column. Were the percentages any different?

**Response**: We propose modifying the text lines 447-450 to the following:

"To calculate the total number concentration of particles with diameters larger than 0.5 µm used in these two parameterizations, we used the SMEAR II APS data for the boundary-layer samples (Fig. 7c, and e). This choice was motivated by the similarities between the size distributions and particle concentrations measured at ground level and in the boundary layer, the fact that the boundary layer was well-mixed during the aircraft measurements, and to investigate if ground-level measurements can be used in parameterizations to predict INP concentrations observed aloft in the boundary layer. For comparison, the INP concentrations predicted by the DeMott et al. (2010) and the Tobo et al. (2013) parameterizations calculated using the aircraft OPS data is shown in Fig. A5. On the other hand, since the free troposphere was characterized by distinct size distributions and particle concentrations, we used the aircraft OPS data to calculate the total number concentration of particles with diameters larger than 0.5 µm in the free troposphere (Fig. 7d, f, and g). For both sample types, the particle concentration was averaged over the sampling time of each sample."

With the following figure in the Appendix:

[Figure]

**Figure A5. Comparison between the INP concentrations observed in the boundary layer and the INP concentrations predicted using the parameterizations from a) DeMott et al. (2010), and b) Tobo et al. (2013) using the aircraft OPS data. DeMott et al. (2010) reproduces 60 % and 32 % of the data points within a factor of 5 and 2, respectively. Tobo et al. (2013) reproduces 49 % and 17 % of the data points within a factor of 5 and 2, respectively. The black solid line represents the 1:1 line while the grey shaded area indicates a range of a factor of 5 from the 1:1 line. The red solid lines show a linear regression fit through the logarithmically transformed data points. The slope of the fit and the number of data points used is shown in each panel.**

Finally, we propose mentioning the very similar percentages obtained line 475:

"Very similar results are obtained when calculating the DeMott et al. (2010) and Tobo et al. (2013) parameterizations using the aircraft OPS data instead of the SMEAR II APS data (Fig. A5), where DeMott et al. (2010) reproduces 60 % and 32 % of the data points within a factor of 5 and 2, respectively, while Tobo et al. (2013) reproduces 49 % and 17 % of the data points within a factor of 5 and 2, respectively. This highlights that both ground-level measurements and aircraft measurements from the boundary layer produce similar parameterization results, which suggests that ground-level measurements are sufficient for predicting INP concentrations aloft in the boundary layer."

Line 476: I think successfully is too strong of a word here. I agree that it performs the best out of all tested parameterizations (which is a nice finding of your study), but you need to take care to qualify and not overstate your conclusion. It's possible that another parameterization out there may fit your data better. The fit line shows the limitations.

**Response**: We propose removing the word "successfully" and rephrasing to:

"[…] among the parameterizations tested here, the Schneider et al. (2021) parameterization performs best at predicting the concentration of INPs in the boundary layer above a Finnish boreal forest environment."

Line 482: This is another sentence that needs qualifying. Yes, strictly speaking, based upon the factor of 5 and 2 percentage of points (which come with uncertainty as INP measurements have large error bars), the parameterization works better for the boundary layer. But visually, comparing Figure 7a and 7b, they look similar. Your statement "Thus the Schneider et al. (2021) parameterization can successfully represent the well-mixed boundary layer, but not the more remote free troposphere where INPs can be more scarce and originate from distant sources," is not convincing to me as the percentages and fit slopes are too similar to warrant a statement this strong. The word "successfully" appears again in the conclusions and abstract.

**Response**: We agree with the referee that the statement should be clarified and we propose rephrasing line 482 to:

"Thus, the Schneider et al. (2021) parameterization performs relatively better at representing the well-mixed boundary layer rather than the more remote free troposphere where INPs can be scarce and originate from distant sources."

In addition, we propose replacing "successfully predicted" line 24 of the abstract and line 655 in the conclusions by "best predicted".

Line 526: The onset difference here mostly looks related to limit of detection.

**Response**: We propose adding the following line 526:

"[…] although the ice onsets of our measurements are approximately 4 °C colder, likely due to differences in the limit of detection."

Figure 8: I think it would be helpful to indicate which studies are from what zone: ground, boundary layer, or free troposphere. This could either be accomplished in the legend with text or by grouping markers in the figure. It is confusing the way it is presented both in the figure and in the text right now. You could also trim the cold end to make the measurements easier to read, as few of yours go much colder than -25 °C. Adding an additional or two free troposphere study would also add value, as it seems that portion is lacking. This would help strengthen or potentially modify your statement in Line 541 saying the free tropospheric measurements fall within the same range as previous measurements. It would be insightful to compare how the free troposphere stacks against other free troposphere studies. Some that may be of use (both ground/airborne studies) would be Conen et al. (2022: https://acp.copernicus.org/articles/22/3433/2022/acp-22-3433-2022.html) Lacher et al. (2018: https://agupubs.onlinelibrary.wiley.com/doi/full/10.1029/2018JD028338) Barry et al. (2021: https://agupubs.onlinelibrary.wiley.com/doi/10.1029/2020JD033752), in addition to the Levin et al. (2019) study I mentioned previously. You don't need to add all of these studies, I just think that it would bolster your figure/argument.

**Response**: We thank the referee for their useful suggestions which have greatly improved the figure. We modified Fig. 8 by grouping the studies by zone (boundary layer, free troposphere, and others) based on color in the figure, and we highlighted these groups in the legend. We also added

the studies from Levin et al. (2019), Conen et al. (2022), and Barry et al. (2021) to provide more comparison to our samples:

[Figure]

**Figure 8: INP concentrations from the present study compared with literature data. The measurements from Sanchez-Marroquin et al. (2021) were conducted in the boundary layer in the South East of the British Isles, while Hartmann et al. (2020) conducted their measurements in the High Arctic boundary layer. For the Twohy et al. (2016) data collected in the western United States, the light blue diamonds represent a filter sampled within the boundary layer (at 1067 m a.g.l.), while the dark orange diamonds represent a filter sampled primarily in the free troposphere (between 897 and 3638 m a.g.l.). For the Levin et al. (2019) study conducted in California, the data represented here correspond to the measurements made in the boundary layer (below 2 km). The measurements from Barry et al. (2021) were conducted between 1300 and 5100 m above sea level in the western United States. The measurements from Conen et al. (2022) were conducted under free troposphere conditions at Jungfraujoch (3580 m above sea level) in the Swiss Alps. The grey band represents the data range given in Petters and Wright (2015) derived from precipitation samples collected around the world. The measurements from Schrod et al. (2017) were conducted between 500 and 2500 m a.g.l. (likely both in the boundary layer and in the free troposphere) over the Eastern Mediterranean.**

We suggest rephrasing lines 524-546:

"Most of the data presented in this study fall within the mid-latitude data range given by Petters and Wright (2015; grey band in **Error! Reference source not found.**) derived from precipitation samples collected around the world, except for the highest INP concentrations measured between -18 and -24 °C in the boundary layer. Part of the data presented in this study also overlap with some of the INP concentrations reported by Schrod et al. (2017) who sampled Saharan dust plumes over the Eastern Mediterranean.

Concerning the boundary layer measurements, most of the INP concentrations presented in this study are higher than concentrations measured in the marine boundary layer in the Arctic during winter (Hartmann et al., 2020), in coastal California during wintertime (Levin et al., 2019) , and above a forested site in the western United States (Twohy et al., 2016). Compared to INP measurements conducted in the South East of the British Isles (Sanchez-Marroquin et al. 2021), the INP concentrations we observed in the boundary layer are about one order of magnitude lower for temperatures above approximately -18 °C, but are within the same order of magnitude for temperatures below approximately -18 °C.

The majority of the INP concentrations measured in the free troposphere in this study fall within the higher range of free tropospheric measurements from Barry et al. (2021) conducted during wildfire events in western US. The INP concentrations from Conen et al. (2022) observed under free troposphere conditions at Jungfraujoch in the Swiss Alps are relatively lower than the concentrations reported here at similar temperatures (≈ -15 °C). In addition, INP concentrations measured in the free troposphere over a forested site in the western United States (Twohy et al., 2016) are about one order of magnitude lower than the average INP concentrations observed in the free troposphere in the present study.

Thus, the INP concentrations measured in the boundary layer and in the free troposphere are mostly higher or within the same range as previous measurements from various regions. These observations illustrate that both the boundary layer and the free troposphere above the Finnish boreal forest are relatively rich in INPs with concentrations comparable to other environments."

Lines 557-559: Is it possible the water negatives were high in this particular sample, thus causing the gap between sample and negative to be too low, causing this junction? Sometimes you can just get unlucky and could also be physical if certain INPs are being inactivated in large volumes of dilution water. The shorter sampling time reason doesn't make a lot of sense, and you should be able to limit particle settling by resuspending the sample before dispensing. Ideally, I would suggest to rerun this sample because it is a part of your case study (maybe pick a lower level of dilution). If there isn't much sample left, you could just run the dilution. However, maybe there isn't any sample left in which case there's nothing you can do. In any case, I would remove the bit about shorter sampling time, because clearly you are still able to get detection to almost -15 °C.

**Response**: There is no remaining sample, and we are not able to re-analyze this sample. We propose rephrasing lines 557-559:

"Note that the discontinuity observed in the free troposphere sample from 17 May 2018 occurs at the dilution step. Possible explanations for this include the inactivation of INPs during dilution, the comparably low amount of sampled aerosol due to the shorter sampling time used for this filter (≈ 45 minutes), or insufficient redispersion of the suspension and the resulting inhomogeneity caused by particle settling (Harrison et al., 2018)."

Section 3.7 General: Overall, I think the case study portion is well done and provides evidence at different angles. I really like the potential cloud processing depletion signal. The only caveat to this section I would mention is that just because the aerosol responds in a certain way doesn't mean the INP, a very small fraction of the total, will respond the same.

**Response**: We thank the referee for their positive feedback. We agree that we should mention the limitations of our analysis and we propose adding the following line 613:

"Furthermore, it is important to stress that, although the analysis of the particle number size distributions and concentrations gives valuable information on the vertical distribution and physical mixing state of the aerosol population, such information cannot necessarily be directly extended to the INPs, which represents a very small and highly variable fraction of the overall aerosol population (DeMott et al., 2010)."

Lines 628-629 and Figure 11c: I am not familiar with FLEXPART, but how certain can you be that there was little time below 200 m if the vertical resolution is 250 m? I say this because the lowest level seems to have an unexpected stripe with little variation. Could this be an artifact? It seems odd to me that there would be little surface influence the whole way. Again, I am no expert here, just pointing out an observation.

**Response**: The sentence line 629 should indeed be modified to 250 m a.g.l. instead of 200 m a.g.l. We updated the manuscript accordingly. The "stripe" visible at the lowest level is not an artifact but simply the result of the low PES values encountered in this layer, as highlighted in the figure below where the PES was simulated with a higher vertical resolution near the surface (for 50, 200 and 250 m) and compared with results obtained at 3000 m:

[Figure]

This figure clearly shows that the PES values at 250 m and below are substantially low compared to the much higher PES values at 3000 m a.g.l., as observed in Fig. 11c.

Line 630: The way it is written is confusing, I think it would be more correct to say that the air spent time in a particular layer.

**Response**: We propose rephrasing to:

"Results show that the air masses spent very little time below 250 m a.g.l., and therefore are less likely to have accumulated surface particles in transit. However, Fig. 11c shows that the airmasses did spend time in the boundary layer, even on the same day as the aircraft measurements."

Conclusions: I think this section can be trimmed down and focused: it is lengthy and carries over some of the issues I noted in the main text.

**Response**: Based also on the comments from Referee 1, we recognize the need to recompose the conclusions. We propose a new text that is included in the response to Referee 1 and in the revised manuscript.

Lines 669-672: I don't agree that you would need longer sampling times in order to do treatments on the suspensions: even 5 mL should give enough leftover volume to do one treatment and could be informative especially on some of your higher signal samples here. Definitely it would not be worthwhile for all of them, since some of them are near the negatives. It would be a good way to confirm the particles are similar with previous work. If you want to leave it for future work, that is fine, but I would suggest to remove/revise the explanation given.

**Response**: We agree with the referee that heat treatments would have been an important additional information in this study. We propose rephrasing the lines 669-672 to:

"Future measurements should include additional analysis of the chemical composition and heat sensitivity of the sampled INPs, in a similar manner to what Hartmann et al. (2020), Hill et al. (2016), and Sanchez-Marroquin et al. (2023) have done."

**Updated figures:**

[Figure]

Figure 1: a) Example of a flight track from Tampere-Pirkkala airport to SMEAR II. The distance from the airport to the station is approximately 60 km. The location of SMEAR II with respect to Northern Europe is given in Fig. A1. b) Instrumental setup viewed from above inside the Cessna 172, described in detail in section 2.1.1.

[Figure]

**Figure A2: Location of SMEAR II and the Tampere-Pirkkala airport with respect to Northern Europe.**

[Figure]

**Figure A2: a) INP concentrations per ml of aerosol solution from the aircraft samples compared to the background signal derived from the handling blank filters collected onboard the aircraft. b) INP concentrations per ml of aerosol solution from the ground-level samples collected at SMEAR II at the same time as the aircraft samples compared to the ground-level handling blank filters. c) INP temperature spectra of all the samples collected during the aircraft measurement campaign together with the ground-level data from Schneider et al. (2021) collected in Hyytiälä during April and May 2018. The error bars represent the statistical as well as the systematic error of the INSEKT assay. More details related to the calculation of these error bars is given in Schneider et al. (2021).**

---

## Author Comment (AC3)

Vertical distribution of ice nucleating particles over the boreal forest of Hyytiälä, Finland

Zoé Brasseur et al.,

This manuscript describes ice nucleation measurements collected in three different atmospheric compartments; ground level, boundary layer, and free troposphere. Boundary layer and free troposphere measurements were made aboard a CESSNA aircraft. The duration of the sampling period ranged from the 20 April until 20 May 2018, and involved 19 separate flights. Ground based measurements took place at the SMEAR II station in Hyytiala. The aircraft was equipped with filter measurements (for INP analysis) and with SMPS and OPS measurements for aerosol size and number concentration.

Several meteorological and remote sensing measurements were used to determine the development of the boundary layer each day. This information was later used to interpret the variability INP concentrations as a function of altitude.

Measurements are compared to previous studies at the same site over a longer period of time. These measurements and the subsequent aerosol measurements are used to develop an updated parameterization that is compared to existing parameterizations from a number of different studies.

There are few studies capable of providing measurements on the vertical profile of INP measurements and therefore this study presents a very unique and valuable data set. The manuscript is well written and the graphic are of very good quality.

**Response**: We thank the referee for their useful comments and suggestions which helped us improve the manuscript. We provide our responses below each individual comment.

**Please find below some general comments and suggestions:**

**Data availability:**

It would be appreciated to provide accessible data to the reviewers (as is stipulated in the policies of ACP).

**Response**: The data will be made available as soon as possible with the following DOI: 10.5281/zenodo.10975295.

**Introduction:**

The presentation and discussion of the parameterizations is an important part of this manuscript. A short discussion on the need for parameterizations in the introduction would be useful, and to present why having multiple variable parameterization's is useful compared to single parameterization's (currently used in cloud models). Can cloud models integrate these more complex parameterizations?

**Response**: We agree with the referee that parameterizations could be mentioned in the introduction. We propose adding the following text after line 50:

"To overcome our lack of knowledge concerning INPs, there is a need for more observations of INPs worldwide. Such measurements are also needed to develop accurate parameterizations, which are an important tool used to constrain heterogeneous ice nucleation predictions in models (e.g., Fletcher et al., 1962; Meyers, 1992; DeMott et al., 2010)."

However, we believe that a discussion related to the different parameterization types and their implementations in models would be out of scope in this manuscript, especially since the comparison with existing parameterizations only represents one section of this manuscript and does not focus on producing new parameterization(s).

**Methodology:**

- Were any samples analyzed immediately after collection and again post freezing. What kind of impact is the freezing cycle expected to have on the ice nucleation activity of aerosol particles?

  **Response**: The freezing of the suspensions of aerosol sampled at ground level at SMEAR II during the HyICE-2018 campaign did not have a significant impact on the measured ice nucleation activity (Kaufmann, 2019; see Fig. 3.13 extracted from the master thesis below). For some samples, we observed a slight reduction of the ice nucleation activity in the higher temperature regime (within the margin of error). It is therefore possible that the freezing of the filters sampled during the aircraft campaign might also have slightly altered the ice nucleation activity in the higher temperature regime. For this reason, the storage of the aerosol filters was kept as short as possible. Warm storage of filters was not an alternative, as it alters the activity especially for biological INPs (Stopelli et al., 2014).

[Figure]

Figure 3.13: **Impact of freezing the aerosol suspensions.** For testing the impact of freezing of the aerosol suspension, several randomly chosen aerosol samples of the HyICE2018 campaign have been analysed directly after producing the aerosol suspension (blue) and a second time after the suspensions have been left in the freezer for several weeks (red).

- Line 189: Can the authors rephrase this sentence. What exactly was passed through the filter and what was the purpose of using this filter. "First, the sampled aerosol particles were washed from the filter membrane into a solution using Milli-Q purified water (18.2 MΩ.cm), which was passed through a 0.1 µm Whatman syringe filter."

  **Response**: We agree that the sentence needs to be clarified and propose rephrasing:

  "First, we used Milli-Q purified water (18.2 MΩ.cm), which was passed through a 0.1 µm Whatman syringe filter to remove possible remaining impurities, to wash the sampled aerosol particles from the filter membrane into a solution."

- What is the volume of liquid used in the sample wells of INSEKT, and how many wells are used? Was it possible to perform multiple runs on the same samples to determine the reproducibility of the filters? What impact of biological aerosol is thought to have on these samples? Schneider et al., performed a heat treatment of the samples, and observed a significant fraction of biological INPs. Do the authors suspect that this might also be the case in these samples, and how would this vary with altitude?

  **Response**: Concerning the analysis with INSEKT, we propose adding the following information to the text line 189:

  "The resulting aerosol suspensions were then analyzed with the INSEKT by pipetting volumes of 50 µL into two 96-well polymerase chain reaction (PCR) plates. Typically, 32 wells were filled with nanopure water while the remaining 160 wells were used to analyze two samples at once. For each sample, 24 wells were filled with the undiluted suspension, 24 wells were filled with a 10 or 15-fold diluted suspension, and 32 wells were filled with a 100 or 225-fold diluted suspension (Schneider et al., 2021)."

  We did not perform multiple runs on the same samples to determine the reproducibility of the analysis.

  In lines 376-379, we make the hypothesis that similar INPs could have been sampled in our study and in Schneider et al. (2021). As mentioned, Schneider et al. (2021) found that the sampled INP population was dominated by heat-labile materials, and one could expect similar features from our ground-level and boundary layer samples. However, since we did not conduct heat tests on the samples collected during the aircraft campaign, we cannot investigate this hypothesis further. We propose adding the following text line 379:

  "It is therefore possible that biogenic particles represent an important fraction of the INPs sampled at ground level and in the boundary layer in this study. A recent study from Maki et al. (2023) showed that airborne microorganisms from forested areas could maintain similar concentrations from ground level up to 500 m under efficient vertical mixing conditions. Since the boundary layer was well-mixed during the aircraft campaign, we can expect that surface biogenic particles had a non-negligible impact on the INPs sampled in the boundary layer.

However, such a hypothesis cannot be confirmed with the data presented in this work, and more measurements such as heat treatment tests (e.g., Hill et al., 2016) would be needed to examine the presence of biogenic INPs in the samples."

- Can the authors provide Fig. A1 in the same units (INP stdL) as the data presented elsewhere. It would be easier for the reader if Figure A1 and A2 are combined in the same figure a) and b).

  **Response**: Since the background correction was done in INP concentration per ml of aerosol solution, we would like to keep this unit in Fig. A1. In addition, following Referee 2's comments on Fig. A1, the figure was modified to indicate which data points were not considered because they did not meet the criterion introduced (INP concentrations at least twice as high as the average background INP concentrations). We agree with the Referee that Fig. A1 and A2 should be combined into one figure with subplots. The updated figure is attached at the end of this document.

- For the air mass backtrajectories, was the impact of airmass history (precipitation, and or cloud activation, from ECMWF calculations) considered in this analysis.

  **Response**: Precipitation for wet deposition (both in cloud and below cloud scavenging) is considered in FLEXPART v10.4. However, since we used a passive air tracer and not aerosol/gases, precipitation does not impact the air mass backtrajectories presented in the manuscript. We propose adding the following information to the methods line 266:

  "The simulations were computed for a passive air tracer for which the wet and dry removal processes have no impact."

**Results:**

- Can aerosol mixing state be inferred from the size distribution measurements? What impact does aerosol mixing state have on INP properties?

  **Response**: Following the definition from Riemer et al. (2019), aerosol mixing state includes both chemical and physical properties of the overall particle population within an aerosol. As such, size distributions only give information on the physical property of the particle population and do not give information on the chemical mixing state of aerosols. Additional measurements of particle properties, such as chemical composition, morphology, optical properties, etc., could tell us more concerning the aerosol mixing state, but such measurements were not made during the aircraft measurements. Furthermore, as discussed in Riemer et al. (2019), the ice nucleation ability of a particle depends on its surface properties, which is in turn strongly tied to the aerosol mixing state. Thus, changes in the aerosol mixing state, for example through chemical aging or coating, can change the ability of the particles to form ice (e.g., Czisco et al., 2009).

  Based on this and Referee 2's comments, we have added the following text line 613:

"In addition, it is important to stress that, although the analysis of the particle number size distributions and concentrations gives valuable information on the vertical distribution and physical mixing state of the aerosol population, such information cannot necessarily be directly extended to the INPs, which represents a very small and highly variable fraction of the overall aerosol population (DeMott et al., 2010)."

- What causes the difference in the onset of freezing between Schneider et al., 2021 and the ground level samples (shown in figure 4). What are the differences between the Schneider sampling set up at SMEAR II and those in this work? The authors say that they were sampling behind a TSP inlet, however Schneider was sampling behind à PM10. What is the inlet height at the ground station?

  **Response**: We suggest that the difference in the onset of freezing between the Schneider et al. (2021) and the ground-level samples is due to the shorter sampling times used for the ground-level samples, as explained lines 309-311. Indeed, the samples presented in Schneider et al. were sampled for 24 hours while the ground-level samples were collected for 3 hours approximately.

  Concerning the sampling set up at SMEAR II, we propose adding the following information lines 176-178:

  "At SMEAR II, the ground-level filter was sampled from a measurement container using a vertical sampling line connected to a total aerosol inlet. The inlet height was approximately 4 m a.g.l. and the average flow rate through the filter was 15 L min$^{-1}$ with an average sampling time of 3 hours."

  In addition, we propose adding some information line 305:

  "Compared to the ground-level samples presented in this study, the samples used in Schneider et al. (2021) were collected from the aerosol cottage (approximately 20 m from the measurement container) using a PM$_{10}$ inlet with an inlet height of approximately 4.6 m a.g.l."

- Figure 4b, would it not be more correct to label this normalized INP / particle number concentration. The axes being labeled Activated fraction, is misleading.

  **Response**: Here we defined the activated fraction as the ratio of INP concentration to the particle number concentration. A similar definition has been used in several previous studies (e.g., Porter et al., 2020; He et al., 2021 and 2023; Iwata et al., 2019) and therefore we believe that we can also use this definition here.

- Can the authors provide more details about the data points in Figure 4 (Median, percentiles etc..).

  **Response**: We propose adding more information line 305:

"In Fig. 4, the data is presented in the form of boxplots calculated for each activation temperature, where the line dividing the boxes in two represents the median value of the distribution, the lower and upper edges of the boxes represent the 25th and 75th percentiles, respectively, the lower and upper whiskers represent the minimum and maximum, respectively, and the outliers are represented as single point markers."

- Since measurements of the full size distribution were available why was the number of INP per aerosol surface area not calculated? This is a pertinent measurement of INP activity and would be possible to put it in context with a number of other studies.

  **Response**: We agree with the reviewer that including the INAS might prove useful to readers and future comparisons, and we therefore propose adding a figure presenting the INAS in the Appendix. We will also add the following text line 334:

  "The ice nucleation active surface site (INAS) densities, calculated as the ratio of the INP concentration to the surface area concentration of particles larger than 300 nm, are presented in Fig. A3."

[Figure]

**Figure A3. a) INAS densities as a function of the activation temperature for all the samples collected during the aircraft measurement campaign together with the ground-level data from Schneider et al. (2021) collected in Hyytiälä from 20 April to 19 May 2018. The INAS densities were calculated by normalizing the INP concentration by the aerosol surface area concentration following the method described in Ullrich et al. (2017) and assuming that each INP triggers the formation of one ice crystal. The aerosol surface area concentration was derived from the size distribution measurements of the particles larger than 300 nm obtained from the OPS and the combined DMPS-APS for the aircraft and the ground-level samples, respectively. b) Number of observations for each sample type as a function of temperature.**

- In the individual plots (shown in Figure 9), we observe very different distributions of INP concentrations as a function of temperature. For example at ground level on the 15th there was a first freezing mode until -18C then a sudden increase in freezing until -20C and then a gradual increase until the end. Whereas in the boundary layer the INP concentrations appear more

homogenous with a consistent rise in INP as a function of temperature. Likewise in the FT measurements there appears to be a similar distribution as the BL. Are these changes in INP spectra a result of dilution steps (as discussed for 9c) or is there additional information that can be extracted? Are there different 'slopes', or freezing modes in the data.

**Response**: The changes in the shapes of the INP temperature spectra are not related to the dilution steps. As illustrated in the figure below, which is an enlargement of Fig. 9a, the changes in slopes do not occur specifically at the dilution steps and thus are not caused by the dilution(s).

[Figure]

It is possible that these differences in the shapes of the INP temperature spectra indicate that different aerosol types dominate the INP populations in the different samples. Similar observations were made in Schneider et al. (2021), but at warmer temperatures (approximately -13 °C). However, without additional analysis of the samples (such as heat tests) or information concerning the chemical composition of the sampled particles, specific conclusion for the variations in the INP temperature spectra is unwarranted. We must leave it to future work to further investigate this matter.

- For the jumps in data points observed for the FT samples collected on the 17th , that the authors state is a result of a dilution step, should the reader focus on the trend in the FT sample after the -20°C of before the -20°C. Are measurements valid in both cases?

  **Response**: As we cannot fully explain the differences measured between the two dilutions, we decided to show both spectra instead of disregarding one of them without good reason. Therefore, we also cannot give guidance on which result is more valid.

- The average INP plots over the full period were illustrated. As expected a high spread in the data points is noted. The authors showed that there was significant changes in weather (Figure 3) over all the flights.

  **Response**: We would like to point out that, with the exception of the first two flights, all aircraft measurements took place in May 2018 during a period with relatively stable meteorological

conditions, as shown in Fig. 3. In addition, while there is some variability in INP concentrations in the samples, this spread is not unusually large (cf Fig. 8 and Fig. A2c) and less than the systematic difference between the free troposphere and the ground-level/boundary-layer samples.

- o Can the variation in the INP concentrations be explained by other variables than aerosol size, such as the temperature and pressure, and also aerosol chemistry (at least from ground based measurements).

  **Response**: All INP concentrations presented in this study were converted to standard conditions in order to eliminate their influence and thus make the samples collected at ground level and in the atmosphere aloft comparable. A direct effect of air temperature and pressure on the INP concentrations at the sampling location is not expected and, to the authors' knowledge, not reported in previous studies. Nevertheless, air temperature can be an indicator for the seasonal cycle and thus can be linked with the boreal forest's biological activity and the concentration of biological INPs, as described in Schneider et al. (2021). However, as mentioned previously, the meteorological conditions were stable in May 2018, hence it is unlikely that the variation in INP concentration is directly linked to meteorology.

  Regarding aerosol chemistry, no measurements of the aerosol chemical composition were conducted onboard the aircraft, and thus we cannot investigate the impact of such parameter on the INP concentrations measured in the boundary layer and in the free troposphere. However, we agree that aerosol chemistry should be investigated in future studies, as mentioned in the conclusions:

  "Future measurements should include additional analysis of the chemical composition and heat sensitivity of the sampled INPs, in a similar manner to what Sanchez-Marroquin et al. (2023), Hartmann et al. (2020) and Hill et al. (2016) have done."

  Concerning ground-level measurements, the link between aerosol chemistry and ground-level INP concentrations is discussed in detail in Schneider et al. (2021). As highlighted in Schneider et al. (2021), there was a clear seasonal transition from winter to spring between the end of March and the beginning of April 2018. However, the aircraft measurements presented here took place after this seasonal transition, when there was less variability in the concentration of biogenic aerosol. Additionally, the focus of this study is not on the ground-level INP measurements but on the aircraft measurements, for which we do not have any aerosol chemistry information.

- o Can aerosol chemistry (other than biogenic sources) be mentioned, at least for the ground based measurements (from measurements available at the SMEAR station), or those measurements already available in the Schneider paper. Was there any variability observed during the sampling period discussed in this manuscript.

  **Response**: Regarding the chemical composition of particles from sources other than biogenic, there is limited data to investigate potential links with the INP concentration

since the only instrument that gives aerosol chemical information is a long time-of-flight aerosol mass spectrometer (L-ToF-AMS) located at ground level in SMEAR II. However, this instrument measures in the size range 75-650 nm, which is generally not considered to be of the highest relevance for ice nucleation in the atmosphere, and only gives information in broad chemical categories including organic, sulfate, nitrate, ammonium, and chloride. Schneider et al. (2021) showed some slight correlation between the mass concentration of non-refractory organic compounds and INP concentrations and suggested that these were linked to the forest biological activity. However, such observations were made over long time scale to study the seasonal cycle of INP concentrations, and correlations might be less visible on shorter time scales such as those covered with the aircraft campaign. As mentioned previously, these ground-level measurements are discussed in Schneider et al. (2021) and since the study presented here focuses on the aircraft measurements conducted aloft (for which we do not have aerosol chemistry measurement), we refrained from investigating ground-level aerosol chemistry.

- **Figure 9**: The authors mention that there were clouds present between 3000 and 4000 m on the flight of the 17th. Was the aircraft sampling inlet adapted to sample could droplets?

  **Response**: The aircraft sampling inlet and setup in general were primarily developed for new particle formation studies (see for example Lampilahti et al., 2021). During the HyICE-2018 campaign, the setup was slightly modified to samples particles on filters in order to analyze their INP content subsequently. Measuring in-situ ice crystals and cloud droplets was not the objective of the campaign, and therefore the aircraft inlet was not adapted for such in-situ measurements.

- **Figure 7**: For the Schneider parameterization (in both the BL and the FT), it seems that there are only a small number of points that are pulling the fit away from the 1:1 line.
  - Do these points correspond to a single flight? If so, is there something particular about this flight?

    **Response**: Assuming that the referee's comment concerns the few data points that are the further away from the grey shaded area (representing a range of a factor of 5 from the 1:1 line) in Fig. 7, where the observed INP concentration are higher than about 10 L$^{-1}$ at temperatures around -20 °C, then no, these points do not correspond to a single flight. For the boundary-layer samples, these points come from three different flights on 14, 15, and 18 May 2018. These three flights had relatively high INP concentrations between -18 and -24 °C which were not reproduced by the Schneider et al. (2021) parameterization. However, the reason behind these elevated INP concentrations remains unclear and is not further explored. For the free troposphere samples, very few points are outside of the grey shaded area representing a range of a factor of 5 from the 1:1 line. The main deviation is observed for the sample collected on 16 May 2018, which corresponds to the highest INP concentrations measured in the free troposphere during the campaign and which is further discussed in section 3.7.

  - Why do the authors not provide a test for the ground based measurements?

**Response**: The focus of this manuscript is primarily on the aircraft measurements conducted at higher altitude, and the ground based measurements are used to bring context and additional information to the vertical profile of INP concentrations. Ground based measurements from HyICE-2018, including their representations by parameterizations, are described in length in the previous studies summarized in the introduction (Paramonov et al. 2020; Schneider et al. 2021; Vogel et al. 2024). Thus, we do not provide additional tests here to avoid repetition, and we prefer focusing on the aircraft INP measurements which were not presented previously.

We propose modifying the text starting line 433 to clarify:

"In Fig. 7a-f, the INP concentrations measured in the boundary layer and in the free troposphere are compared to INP concentrations predicted by three existing parameterizations from Schneider et al. (2021), DeMott et al. (2010), and Tobo et al. (2013), which are presented in Table 1. This section focuses on the aircraft INP measurements conducted at altitude, and a detailed comparison between parameterizations and INP concentrations from ground-based filter measurements similar to those presented here can be found in Schneider et al. (2021)."

- In figure 7: Is it possible to apply this newly developed parameterization's to the ground based, boundary layer data, and also to Schneiders data. It would be interesting to determine if this parametrization can be used in other environments or if it is only suitable for FT measurements.

**Response**: The adjusted parameterization presented in Fig. 7g is not a newly developed parameterization, but simply the parameterization from Tobo et al. (2013) with adjusted coefficients. To avoid confusion, we modified the legend of Fig. 7g to "Adjusted Tobo et al. (2013)" and clarified the method used to adjust the parameterization (see comment below). Considering that the parameterization from Tobo et al. (2013) was adjusted to better fit the free troposphere data, we do not see value in testing it on ground or boundary layer measurements. In addition, as we stress in the manuscript lines 505-507, this adjusted parameterization should be used with caution as it was derived for a very limited number of observations (only 12 filters samples/119 data points), and we do not claim that it is suitable for all free troposphere measurements or other environments.

- In Table 1, it would be useful to also include the fitting values of the updated parameterization included here. Those that are listed at the end of the paragraph.

**Response**: We propose modifying Table 1 to the following:

| Reference | Temperature range | Equation | Input parameters |
|---|---|---|---|
| Schneider et al. (2021) | -25 to -12 °C | $n_{INP} = 0.1 \cdot exp(a1 \cdot T_{amb} + a2) \cdot exp(b1 \cdot T + b2)$ with $a1$ = 0.074 K$^{-1}$, $a2$ = -18, $b1$ = -0.504 K$^{-1}$, and $b2$=127 | Ground-level ambient air temperature $T_{amb}$ (K) Activation temperature $T$ (K) |
| DeMott et al. (2010) | -35 to -9 °C | $n_{INP} = a(273.16 - T)^b (n_{AP,>0.5\ \mu m})^{(c(273.16-T)+d)}$ with $a$ = 0.0000594, $b$ = 3.33, $c$ = 0.0264, and $d$ = 0.0033 | Number concentration of particles with diameters larger than 0.5 μm $n_{AP,>0.5\ \mu m}$ (cm$^{-3}$) Activation temperature $T$ (K) |
| Tobo et al. (2013) | -34 to -9 °C | $n_{INP} = exp(\gamma(273.16 - T) + \delta)(n_{AP,>0.5\ \mu m})^{(\alpha(273.16-T)+\beta)}$ with $\gamma$ = 0.414, $\delta$ = -6.671, $\alpha$ = -0.074, and $\beta$ = 3.8 | |

| Adjusted Tobo et al. (2013) | -34 to -9 °C | $n_{INP} = exp(\gamma(273.16 - T) + \delta)(n_{AP,>0.5\ \mu m})^{(\alpha(273.16-T)+\beta)}$ with $\gamma = 0.7408$, $\delta = $ -16.0788, $\alpha = 0.2746$, and $\beta = $ -3.3184 |
|---|---|---|

- In the development of the parametrization, what methods were used to find the optimum values for the coefficients? Were these calculated only using the FT samples?

**Response**: The adjusted coefficients were indeed calculated using only the free troposphere samples, as mentioned lines 500-501. We propose rephrasing lines 499-501 and adding some information to clarify how such adjusted coefficients were obtained:

[revised manuscript text omitted]

Stopelli, E., Conen, F., Zimmermann, L., Alewell, C., and Morris, C. E.: Freezing nucleation apparatus puts new slant on study of biological ice nucleators in precipitation, Atmos. Meas. Tech., 7, 129–134, https://doi.org/10.5194/amt-7-129-2014, 2014.)

Tobo, Y., Prenni, A. J., DeMott, P. J., Huffman, J. A., McCluskey, C. S., Tian, G., Pöhlker, C., Pöschl, U., and Kreidenweis, S. M.: Biological aerosol particles as a key determinant of ice nuclei populations in a forest ecosystem, J. Geophys. Res. Atmos., 118, 10,100-10,110, https://doi.org/10.1002/jgrd.50801, 2013.

---

## Author Response (AR1)

The manuscript by Brasseur et al focusses on the vertical distribution of INP over boreal forest in Finland and has been carried out by aircraft measurements during spring 2018 above Hyytiälä. The manuscript is well-written and includes new and important data. The topic is important for atmospheric scientists and might be published in ACP after some minor corrections.

**Response**: We thank the referee for their useful comments which helped improve the manuscript. We include our responses below in the context of individual comments.

**Comments**:**

Introduction: The topic has been well introduced and the relevant new literature has been cited. However, the authors might mention their own paper Vogel et al. which is under discussion as well. Also, the older literature at similar locations deserves attention (e.g. Prenni 2013). The authors mention the ice nucleation temperature of desert dust being below -15°C but I miss a statement regarding the biological INPs and INMs which are below -2°C depending on the origin. In this regards the authors might also mention that dust could be a carrier, coated with bio INMs, which turns them into bio INPs.

**Response**: We propose adding a mention to the Vogel et al. (2024) paper line 112:

"Finally, Vogel et al. (2024) presented ground-based measurements conducted with the Portable Ice Nucleation Experiment (PINE) below -24 °C and found moderate correlations between INP concentrations and concentrations of particles larger than 0.5  $\mu$ m as well as concentrations of fluorescent aerosol particles, hinting at a possible biological source of INPs active below -24 °C."

In addition, we propose mentioning previous studies conducted at similar locations (forested environments) line 93:

"These results agree well with previous studies conducted in similar forested environments. Prenni et al. (2009) for example showed that INP concentrations and abundance in a pristine rainforest of the Amazon basin could be partly explained by local emissions of biological particles. Huffman et al. (2013), who performed measurement in a semi-arid pine forest of North America, found a strong correlation between fluorescent biological particles and INPs during rain events. Similarly, results presented in Prenni et al. (2013) suggest that biological particles represent a significant portion of rain-generated INPs measured at a forested site in Colorado. Tobo et al. (2013) conducted measurements in a midlatitude ponderosa pine forest ecosystem in Colorado and found significant correlations between INP concentrations and the concentration of ambient fluorescent biological particles played an important role as INPs for temperatures warmer than -22 °C, especially during rainfall events. However, all these observations were carried out at ground level and did not examine the transport of such INPs to higher altitudes."

Concerning biological INPs, we propose modifying lines 47-50 to:

"Biological aerosols are considered another widely present type of INPs (O'Sullivan et al., 2015; O'Sullivan et al., 2018; Wex et al., 2019;). Although their global emissions are lower than dust, they can form ice at relatively warmer temperatures depending on the nature of the bioaerosols (Després et al., 2012). For example, the bacteria *Pseudomonas Syringae* is a very efficient INP at temperatures as warm as -2 °C (Maki et al., 1974; Joly et al., 2013). In addition, biological particles, including bacteria, have been found in dust aerosols, possibly enhancing their ice nucleation activity (Conen et al., 2011; Meola et al., 2015; Barr et al., 2023)."

Methods: The description is very detailed. However, I have remarks regarding the figures:

Figure 1a: The map is very pale and the scripts are small. I would recommend to add a general map where the location on the European continent is documented.

**Response**: Following the referees' comments, Fig. 1a was updated to a 3D plot to highlight the flight tracks more clearly. A figure with the location of the measurement site with respect to the European continent was added in the Appendix. The updated figures are attached at the end of this document.

Figure 2: The color code is missing. The x-axis might be labeled "day time" or "position of the sun"

**Response**: We added a legend with the color code and a label for the x-axis. The updated figure is attached at the end of this document.

Results: Figure 3: Please explain the calculation of the TKE dissipation rate from the Doppler lidar in more detail.

**Response**: Thank you for your comment. We propose modifying the text lines 245-251 to the following:

"After the flights, data from a Halo Photonics Stream Line scanning Doppler lidar located at SMEAR II was used to estimate the limit between the boundary layer and the free troposphere, for comparison with the aircraft measurements. The Halo Doppler lidar was configured with vertically-pointing stare and conical scans (i.e., with velocity azimuth display (VAD) scans at 30 ° elevation angle) repeating every 30 minutes. Additional scans during the 30-minute scan cycle were not used in this study. The range resolution of the lidar is 30 m, with a minimum range of 90 m a.g.l.. More details on the Doppler lidar at SMEAR II can be found in e.g., Hellén et al. (2018). The data was post-processed following Vakkari et al. (2019): horizontal winds were retrieved from the VAD scans and the variance of vertical wind velocity was calculated from 12 consecutive vertical stare measurements. The instrumental noise contribution to the observed variance of vertical wind velocity was estimated from the post-processed signal-to-noise-ratio according to Pearson et al. (2009) and subtracted before calculating turbulent kinetic energy (TKE) dissipation rate profiles according to the method by O'Connor et al. (2010)."

Figure 5a: add ticks on the y-axis for every order of magnitude.

**Response**: We added ticks for every order of magnitude on the y-axis. We attach the updated figure at the end of this document.

Figure 6a: add a size bar, explain the color code in more detail

**Response**: We added a title to the legend and modified the figure caption to improve clarity regarding the color code. We attach the updated figure at the end of this document. However, we would rather not add a scale bar in Fig. 6a as it may be inaccurate due to the distortions that can occur with map projections at this scale. We believe that the gridlines representing the latitudes and longitudes should be enough to provide context to the map used.

Figure 7: add ticks on the x- and y-axes for every order of magnitude

**Response**: We added ticks for every order of magnitude on the x and y axes. We attach the updated figure at the end of this document.

Figure 11a and b: add size bar. Better explain PES.

**Response**: As in Fig. 6a, we would prefer not to add scale bars to Fig. 11a and b since adding a straight line to represent the scale would be inaccurate due to distortions in the projection. Concerning the PES, we propose adding the following text to the methods line 264:

"FLEXPART was used to calculate the potential emission sensitivity (PES) fields, where PES is the response function of a source-receptor relationship which estimates the potential source contributions for a given receptor (in this case the measurement site SMEAR II). The PES is therefore proportional to the residence time of the air mass in a specific grid cell, and it was calculated in units of seconds. High values of PES indicate source regions where emissions are likely to significantly impact the tracer concentrations at the receptor (Seibert and Frank, 2004; Stohl et al., 2005; Pisso et al., 2019)."

Conclusion: The conclusion reads more like a summary. I miss a real discussion and conclusion from the results along these questions: What do we learn regarding the emission of INP and their transport from the forest ecosystem, above canopy, into the free troposphere? Can you speculate about the transport mechanisms? How do wind and humidity influence these potential processes?

**Response**: Following the comments from Referees 1 and 2, we have shortened and focused the conclusions. Here is the updated text:

"In this study, we present the first aircraft measurements of INP concentrations above the Finnish boreal forest, and we shed new light on the vertical distribution of INPs above this environment. We found that local surface particles were transported and mixed within the boundary layer aloft through convective mixing, resulting in similar INP concentrations and activated fractions observed at ground level and in the boundary layer. INP concentrations and activated fractions measured in the boundary layer were within the same order of magnitude as those reported by Schneider et al. (2021) for the same period and were best predicted by the parameterization developed in the same study. This further suggested that INPs sampled in the boundary layer

primarily originated from the local boreal forest environment rather than long-range transported particles. Although the identity of the INPs sampled in the boundary layer, and whether or not they are dominated by biogenic aerosol similarly to what Schneider et al. (2021) found, has yet to be confirmed, our results suggest that the Finnish boreal forest is the main source of INPs observed in the boundary layer. Future measurements should include additional analysis of the chemical composition and heat sensitivity of the sampled INPs, in a similar manner to what Sanchez-Marroquin et al. (2023), Hartmann et al. (2020) and Hill et al. (2016) have done previously.

On the other hand, much lower INP concentrations were observed in the free troposphere. The distinct particle number size distributions observed there indicated that different aerosol and INP populations were encountered in the free troposphere and that local surface particles have a weaker influence at these altitudes. The free tropospheric INPs likely resulted from long-range transported particles from different sources, although the analysis of the airmass backward trajectories in the free troposphere did not yield conclusive results due to the limited number of observations. Additional measurements are needed to draw conclusions on the influence of the air mass origin(s) on the INP concentrations and identify the source(s) of INPs observed in the free troposphere.

However, one event was observed where INP concentrations and activated fractions measured in the free troposphere were higher and within the same order of magnitude as the concentrations observed at ground level and in the boundary layer. Analysis indicated that the air mass sampled in the free troposphere during this flight was influenced by the boundary layer. Although the exact transport mechanism remains unclear, it is possible that particles and INPs were transported to the free troposphere via boundary layer ventilation, likely caused by convection, turbulent mixing across the capping inversion, or upwards vertical motions of large scale weather systems (e.g., Donnell et al., 2001; Agustí-Panareda et al., 2005). Overall, this finding is of particular importance since INPs in the free troposphere can have longer lifetimes and travel farther, and can therefore impact cloud formation on a regional or global scale."

**Updated figures:**

Figure 1: a) Example of a flight track from Tampere-Pirkkala airport to SMEAR II. The distance from the airport to the station is approximately 60 km. The location of SMEAR II with respect to Northern Europe is given in Fig. A1. b) Schematic of the instrumental setup viewed from above inside the Cessna 172, described in detail in Section 2.1.1.

Figure 2: Schematic diagram of the boundary layer diurnal development adapted from Stull (2017) and Lampilahti et al. (2021), and overlaid with an example flight profile. The actual layer heights may vary from the values depicted on the vertical axis.